# Capturing carbon dioxide from air with charged-sorbents

Huaiguang Li[1,2], Mary E. Zick[3], Teedhat Trisukhon[1], Matteo Signorile[4], Xinyu Liu[1], Helen Eastmond[1], Shivani Sharma[1], Tristan L. Spreng[1], Jack Taylor[1], Jamie W. Gittins[1], Cavan Farrow[1], S. Alexandra Lim[3], Valentina Crocellà[4], Phillip J. Milner[3] & Alexander C. Forse[1 ✉]

Emissions reduction and greenhouse gas removal from the atmosphere are both necessary to achieve net-zero emissions and limit climate change[1]. There is thus a need for improved sorbents for the capture of carbon dioxide from the atmosphere, a process known as direct air capture. In particular, low-cost materials that can be regenerated at low temperatures would overcome the limitations of current technologies. In this work, we introduce a new class of designer sorbent materials known as 'charged-sorbents'. These materials are prepared through a battery-like charging process that accumulates ions in the pores of low-cost activated carbons, with the inserted ions then serving as sites for carbon dioxide adsorption. We use our charging process to accumulate reactive hydroxide ions in the pores of a carbon electrode, and find that the resulting sorbent material can rapidly capture carbon dioxide from ambient air by means of (bi)carbonate formation. Unlike traditional bulk carbonates, charged-sorbent regeneration can be achieved at low temperatures (90–100 °C) and the sorbent's conductive nature permits direct Joule heating regeneration[2,3] using renewable electricity. Given their highly tailorable pore environments and low cost, we anticipate that charged-sorbents will find numerous potential applications in chemical separations, catalysis and beyond.

Hydroxide-based scrubbers are among the most promising for direct air capture (DAC) of carbon dioxide[4,5]. Industrially mature approaches use aqueous KOH solutions[6] or solid calcium hydroxide[7] as the sorbent, but high-energy regeneration steps at 900 °C and the use of natural gas are often required[6]. These high regeneration temperatures arise from the significant lattice energy of the formed carbonate materials and contribute greatly to the costs of running a DAC process. An alternative approach that can significantly reduce regeneration temperatures is to disperse hydroxides in a porous material or polymer matrix[8,9]. For example, hydroxide-functionalized metal-organic frameworks have achieved promising DAC performance with much lower regeneration temperatures (roughly 100 °C), but these materials suffer from limited stabilities and high sorbent costs[10–12]. Motivated by these challenges, we sought a low-cost and robust hydroxide material that could combine DAC with low-temperature regeneration. We proposed that (1) new DAC sorbents could be synthesized by electrochemically inserting reactive hydroxide ions into a porous carbon electrode[13–16], and that (2) the resulting electrically conductive sorbents could be heated using rapid Joule heating (also known as resistive heating) without a secondary conductive support[2,3].

## Preparation and characterization of charged-sorbents

The preparation of charged-sorbents is based on the charging of an electrochemical energy storage device (Fig. 1). During charging, electrolyte ions accumulate in the pores of a conductive porous carbon electrode; for example, anions accumulate in the electrode when charging positively (Fig. 1, step 1). After completing the charging process, this electrode is removed from the cell and is washed and dried to remove residual electrolyte and solvent to yield a charged-sorbent material. Our hypothesis is that the accumulated ions in the porous electrode can then serve as active sites for an adsorption process such as $CO_2$ capture. Different electrode materials, electrolyte ions and solvents can be selected, allowing for the preparation of tailored sorbents for different applications.

Using this approach, we targeted a hydroxide-functionalized porous carbon as a new sorbent for DAC. This was achieved by using inexpensive activated carbon cloth as the electrode (Extended Data Fig. 1) and a 6 M KOH(aq.) solution (Fig. 2a). The activated carbon cloth was positively charged by applying a potential of +0.565 V versus standard hydrogen electrode (SHE) for 4 h, thereby accumulating reactive hydroxide ions within the carbon pores. Both capacitive and faradaic currents were observed during charging, suggesting that hydroxides are accumulated through both electric double-layer formation and oxidation of surface functional groups (Extended Data Fig. 2a,b and Extended Data Table 1). Following charging, the electrode was removed from the cell, and was washed to remove excess KOH and minimize salt formation on the surface of the carbon cloth (Extended Data Fig. 1). Finally, the material was dried to yield a positively charged-sorbent (PCS) bearing hydroxide ions, referred to as PCS-OH (Fig. 2a). Powder X-ray diffraction analysis

[1]Yusuf Hamied Department of Chemistry, University of Cambridge, Cambridge, UK. [2]School of Science and Engineering, The Chinese University of Hong Kong, Shenzhen, China. [3]Department of Chemistry and Chemical Biology, Cornell University, Ithaca, NY, USA. [4]Chemistry Department, NIS and INSTM Reference Centre, University of Torino, Torino, Italy. ✉e-mail: acf50@cam.ac.uk

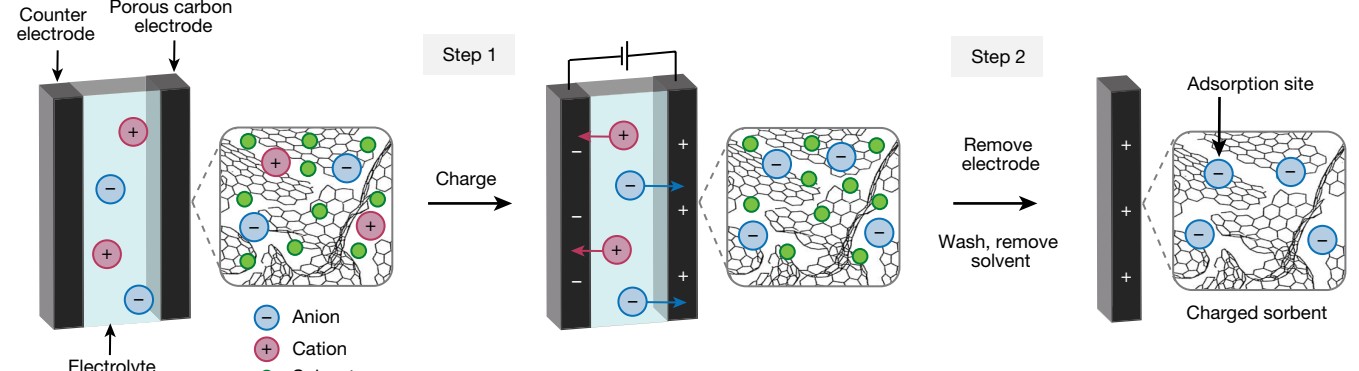

**Fig. 1 | The preparation of charged-sorbents.** A porous carbon electrode is charged in an electrochemical cell (step 1). The electrodes are removed from the cell, washed with deionized water and evacuated to remove solvent molecules (shown as green circles) to yield charged-sorbents (step 2). The activated carbon schematic is adapted from ref. 33, Springer Nature America.

of PCS-OH showed no reflections from crystalline KOH or related products, suggesting that the hydroxide ions are incorporated primarily within the nanometre-sized pores of PCS-OH (Extended Data Fig. 2c). Finally, the counterpart negatively charged-sorbent was prepared as a control sample by applying a potential of −0.235 V versus SHE for 4 h, to yield a negatively charged-sorbent replete with potassium ions (NCS-K).

Evidence for the incorporation of hydroxide ions in the pores of PCS-OH was obtained from $^1H$ solid-state nuclear magnetic resonance (NMR) measurements (Fig. 2b). For PCS-OH, a strong resonance was observed at roughly 1.5 ppm, corresponding to $OH^-$ species present in the pores. This assignment was supported by control measurements on the as-purchased activated carbon cloth ('blank cloth'), a carbon cloth that was soaked in the electrolyte but not charged ('soaked cloth') and the negatively charged carbon cloth sample NCS-K, which all showed much weaker $^1H$ resonances assigned to small amounts of residual $H_2O$ or $OH^-$ species. The more positive $^1H$ chemical shift in PCS-OH probably arises from differences in the ring-current effects from the charged aromatic carbon surfaces[17,18]. Further support for the accumulation of $OH^-$ species in PCS-OH was provided by titration experiments, in which 1.2 mmol $g^{-1}$ of HCl was required to neutralize PCS-OH, compared to 0.2 mmol $g^{-1}$ for NCS-K (Extended Data Fig. 2d). Combustion analysis also supports an increased amount of hydrogen in PCS-OH compared to the controls (Extended Data Fig. 2e), consistent with the charge-driven accumulation of $OH^-$ species in the electrode. Finally, the Brunauer–Emmett–Teller (BET) surface area of PCS-OH was 920 m $g^{-1}$, 20% lower than the blank cloth (1,175 $m^2 g^{-1}$) (Extended Data Fig. 3a,b). We propose that the 20% reduction in surface area for PCS-OH compared to the blank cloth arises from two factors: (1) the accumulation of hydroxide ions in the pores reducing the accessible pore volume, and (2) the electrochemical oxidation of functional groups in the carbon (Extended Data Fig. 2a,b, Extended Data Table 1 and the section on 'Preparation of charged-sorbents') disrupting the carbon's overall porosity. Despite the small decrease in the material surface area, PCS-OH remains highly porous (Extended Data Fig. 3c), suggesting that the accumulated hydroxide ions should be accessible as reactive sites for $CO_2$ adsorption.

$CO_2$ adsorption isotherm measurements (Fig. 2c,d,e) showed enhanced $CO_2$ capture at low pressures for PCS-OH relative to the control samples. This enhanced low-pressure uptake is typical of hydroxide-functionalized sorbents[10,11], supporting $CO_2$ chemisorption by the accumulated hydroxide ions. PCS-OH showed increased $CO_2$ uptake at pressures relevant to DAC. At 0.4 mbar and 25 °C, the $CO_2$ capacity for PCS-OH is 0.26 ± 0.06 mmol $g^{-1}$ (ten independent samples, Extended Data Fig. 4), which is significantly larger than the capacities of the control samples under these conditions (Fig. 2d,e). Experiments

on samples synthesized using less-positive charging potentials and shorter charging times gave smaller $CO_2$ uptake at low pressures due to the incorporation of fewer hydroxide ions (Extended Data Figs. 2d and 5a,b). PCS-OH also showed enhanced low-pressure $CO_2$ uptake compared to a 'dripped cloth' sample, which was preparing by adapting an approach from the literature for impregnating activated carbon with metal hydroxide solutions (Fig. 2e, Extended Data Fig. 5c,d and Methods)[13]. Finally, our charged-sorbents are highly tuneable materials because the electrode (and electrolyte) can readily be varied in the synthesis. As a second example, powdered YP80F activated carbon was first prepared into a free-standing electrode film before synthesizing a charged-sorbent referred to as PCS-OH (YP80F) (Extended Data Fig. 5e and Methods). Enhanced low-pressure $CO_2$ uptake was again observed, demonstrating the generality of the charged-sorbent approach (Extended Data Fig. 5f,g).

To investigate the nature of $CO_2$ sorption in PCS-OH we performed microcalorimetry tests that allowed the direct quantification of the heat released during $CO_2$ uptake measurements (Fig. 2f). For the blank carbon cloth control, the measured $CO_2$ adsorption heat is between −28 and −20 kJ $mol^{-1}$, consistent with $CO_2$ physisorption[19]. A large increase in the adsorption heat is observed for PCS-OH relative to the blank carbon, consistent with $CO_2$ chemisorption. The measured adsorption heats gradually decrease from a value of −137 kJ $mol^{-1}$ at zero coverage (extrapolated value) to −33 kJ $mol^{-1}$ at a coverage of 0.8 mmol $g^{-1}$ indicative of bicarbonate formation at a distribution of hydroxide sites. The adsorption heats compare well with previous reports for bicarbonate formation in porous materials and metal oxides[10,11,20], lending support that the electrochemically inserted hydroxides in PCS-OH serve as $CO_2$ chemisorption sites to enhance low-pressure uptake.

Further to the promising gas sorption and calorimetry results, thermogravimetric analysis (TGA) measurements (using a heated furnace) indicate that PCS-OH has good thermal and oxidative stability. The sorbent was heated to 150 °C under flowing dry air at atmospheric pressure for 12 h. Dry, pure $CO_2$ adsorption isobars before and after air exposure showed very similar capacities and uptake kinetics, confirming the oxidative stability of the material under these conditions (Fig. 2g). This promising oxidative stability is in contrast to the behaviour of amine-based sorbents, which typically show poor oxidative stability, a recurring challenge for their application for DAC[21,22]. We do, however, note that 10% of its capacity was lost after sample storage for 14 months (Extended Data Fig. 6a). As a further stability assay before DAC tests, PCS-OH was subjected to 150 adsorption and desorption cycles using TGA under concentrated $CO_2$ conditions (30% $CO_2$ in $N_2$) (Fig. 2h and Extended Data Fig. 6b). Consistent with the

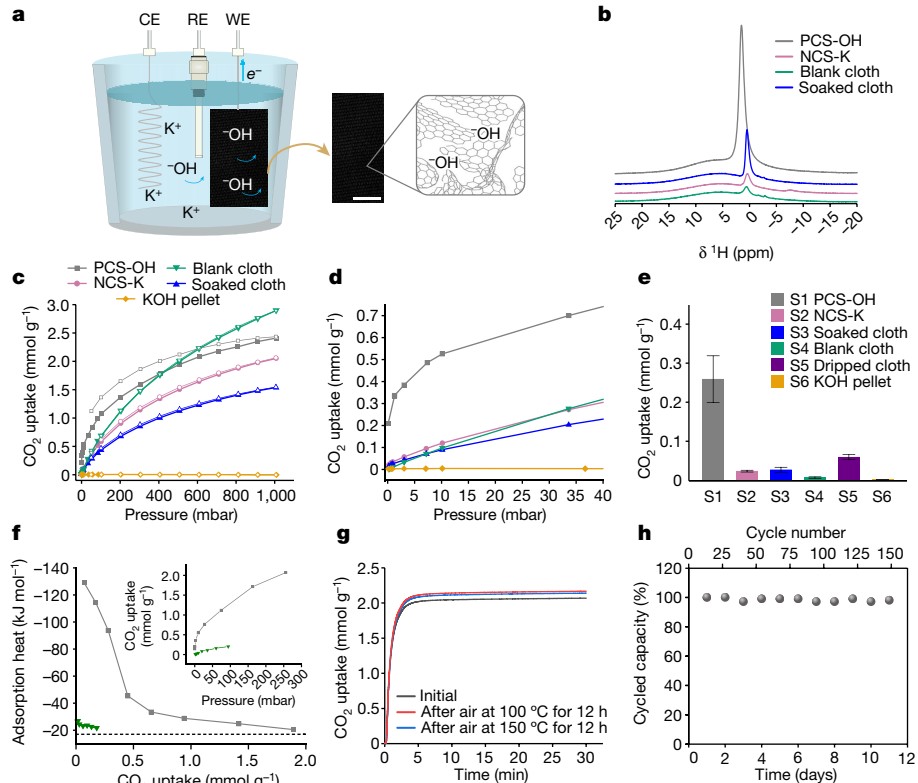

**Fig. 2 | Preparation of hydroxide charged-sorbents and CO₂ sorption.**
**a**, Scheme of charging activated carbon fabric ACC-5092-10 cloth in 6 M KOH through a three-electrode configuration. Scale bar, 0.5 cm. **b**, ¹H solid-state NMR (9.4 T) spectra of PCS-OH and control samples, acquired at a magic angle spinning (MAS) rate of 12.5 kHz. **c**, CO₂ adsorption (filled data points) and desorption (hollow data points) isotherms of PCS-OH and control samples at 25 °C. **d**, Low-pressure region of the CO₂ adsorption isotherms from **c**. **e**, CO₂ uptake of PCS-OH and control samples at 0.4 mbar and 25 °C. Standard deviations are calculated from ten independent samples for PCS-OH, and three independent samples otherwise. **f**, Adsorption microcalorimetry measurements of the differential molar adsorption heats curves related to the adsorption of CO₂ at 30 °C on PCS-OH (grey) and blank cloth (green).

The dotted horizontal line represents the standard molar enthalpy of liquefaction of CO₂ at 30 °C of −17 kJ mol⁻¹. The inset shows volumetric isotherms obtained by performing CO₂ adsorption at 30 °C on PCS-OH (grey) and blank cloth (green) with the volumetric line coupled to the microcalorimeter. **g**, Dry, pure CO₂ uptake curves at 40 °C and 1 bar CO₂ for PCS-OH after activation under flowing dry N₂ at 130 °C for 1 h (grey curve), after exposure to flowing dry air (roughly 21% O₂ in N₂) at 100 °C for 12 h (red curve), and after exposure to flowing dry air (roughly 21% O₂ in N₂) at 150 °C for 12 h (blue curve). **h**, Cycling capacities for 150 adsorption–desorption cycles for the PCS-OH in a simulated temperature–pressure swing adsorption process (Extended Data Fig. 6b). Adsorption 30 °C, 20 min; Desorption 100 °C, 20 min; dry 30% CO₂ in N₂ was used for both the adsorption and desorption steps.

thermal stability test, PCS-OH showed a stable cycling capacity (assuming negligible N₂ uptake), with minimal performance loss after 150 cycles. Low regeneration temperatures of roughly 100 °C can be used (Extended Data Fig. 6c), supporting the idea that dispersal of the reactive hydroxides in the carbon pore network prevents the formation of stable bulk carbonates. Overall, these data support that PCS-OH shows remarkable stability and promising affinity towards CO₂ at pressures relevant to DAC.

## CO₂ capture mechanism

To further investigate the mechanistic pathway responsible for strong CO₂ binding in PCS-OH, we collected ¹³C solid-state NMR spectra for PCS-OH and the control samples after dosing with ¹³CO₂ gas at 0.9 bar (Fig. 3a). All samples showed strong resonances at 119 ppm and weaker resonances at 125 ppm, which are assigned to physisorbed CO₂ in the carbon nanopores and free ¹³CO₂ gas, respectively[23]. The chemical shift difference between these species is consistent with a ring-current shielding for the in-pore physisorbed species, with the shift difference of −6 ppm similar to a recent NMR study on other carbon cloths[24,25]. In contrast to the controls, the spectrum of PCS-OH showed an extra resonance at 156.0 ppm, assigned to chemisorbed (bi)carbonate species (Fig. 3b)[26–29]. Note that carbonate and bicarbonate species are

often in fast exchange on the NMR timescale, so we cannot readily discriminate between these species[30]. Regardless, observation of this chemisorption resonance provides strong evidence that the hydroxide sites incorporated through our electrochemical synthesis chemically react with CO₂, and corroborate the findings from microcalorimetry (Fig. 2f). The chemical shift of 156.0 ppm is similar to bulk potassium bicarbonate (161.4 ppm, Extended Data Fig. 7b), with the smaller value in PCS-OH partly arising from ring-current effects and supporting that the (bi)carbonate species reside in the carbon nanopores (Fig. 3b). Quantitative measurements showed that the amount of physisorbed CO₂ in the four sorbents was similar (around 1 mmol g⁻¹), whereas only PCS-OH chemisorbed CO₂ (0.95 mmol g⁻¹ at 0.94 bar) (Fig. 3c). The observation of 0.95 mmol g⁻¹ of chemisorption aligns with the microcalorimetry measurements, which show a decrease in adsorption heat below −30 kJ mol⁻¹ once this CO₂ loading is reached.

The NMR results further allow us to estimate a lower limit for the hydroxide content in PCS-OH of 0.95 mmol g⁻¹ (assuming a 1:1 reactivity of CO₂ and hydroxide), which is comparable to the value of 1.2 mmol g⁻¹ from titration (Extended Data Fig. 2d). This further leads to an estimated molecular formula for PCS-OH of (OH)C₈₆. Similar NMR spectroscopy experiments on PCS-OH prepared at a lower charging potential (0.365 V versus SHE, instead of our optimized value of 0.565 V versus SHE) provided a much lower limiting hydroxide content of 0.38 mmol g⁻¹ and

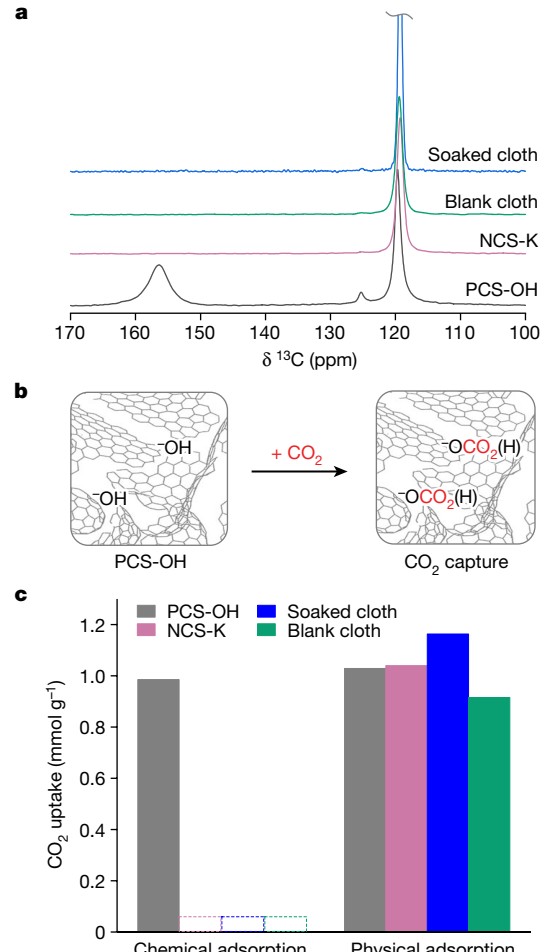

**Fig. 3 | CO₂ binding mechanism from solid-state NMR experiments.**
**a**, Quantitative $^{13}C$ solid-state NMR spectra of sorbents with $^{13}CO_2$ gas dosing at the pressure of 0.9 bar, acquired at a MAS rate of 12.5 kHz (see Extended Data Fig. 7a for a control experiment with no $CO_2$ dosing). **b**, Proposed mechanism for $CO_2$ capture by positively (hydroxide) charged-sorbent. Schematic adapted from ref. 33, Springer Nature America. **c**, $CO_2$ uptake of sorbents by means of chemical and physical adsorption calculated from resonances at 156 and 119 ppm of **a**, respectively.

an estimated molecular formula of $(OH)C_{218}$ (Extended Data Fig. 7c). The lower hydroxide content of samples prepared in this way led to lower $CO_2$ uptakes at low pressures (Extended Data Fig. 5a), supporting a positive correlation between hydroxide content and low-pressure $CO_2$ uptake. Overall, the NMR spectroscopy experiments strongly support that reactive hydroxide ions can be installed in porous carbon through our electrochemical synthesis to achieve greatly enhanced reversible $CO_2$ sorption.

## Demonstration of DAC and regeneration by Joule heating

The promising low-pressure $CO_2$ uptake of PCS-OH through chemisorption motivated DAC tests. DAC performance was initially evaluated under simulated dry air with 400 ppm $CO_2$ at 30 °C, with desorption conducted under 100% $N_2$ at 130 °C. These measurements showed a $CO_2$ capacity of roughly 0.2 mmol g$^{-1}$, similar to the isotherm measurements, which was stable over repeated adsorption and desorption cycles (Fig. 4a). PCS-OH can still do DAC (Extended Data Fig. 8a) at comparable capacity after 14 months, despite the small capacity lost seen under pure $CO_2$ conditions. The DAC kinetics for PCS-OH are also promising,

with a comparable $CO_2$ capture rate to several benchmark sorbents, albeit with a lower saturation capacity (Extended Data Fig. 8b,c).

As a more realistic DAC test, freshly activated PCS-OH and three control samples were subjected to ambient air (37% relative humidity, RH) in a sealed container equipped with a $CO_2$ sensor. A large decrease in $CO_2$ concentration was observed for PCS-OH, with a decrease from 500 ppm to around 25 ppm (a 475 ppm decrease) in 25 min. By contrast, the $CO_2$ concentration decreased much less for the control samples (Fig. 4b). These measurements strongly support that our electrochemical synthesis enhances carbon capture at low partial pressures and enables DAC.

The electrically conductive nature of our PCS-OH sorbent opens the door to regeneration by direct Joule heating[26], a strategy that leads to rapid regeneration times compared to traditional heating methods[2]. In contrast to previous approaches that required the use of electrically conductive supports[2], here we directly attach electrodes to our conductive sorbent for Joule heating (Fig. 4c). A d.c. voltage of 7–8 V was applied across a piece of PCS-OH (2 × 1 cm), resulting in the rapid heating of the material to roughly 90 °C within roughly 1 min. Solid-state NMR experiments on samples predosed with $^{13}CO_2$ gas showed that Joule heating led to the complete release of adsorbed $CO_2$ species in this time (Fig. 4d,e).

We then carried out proof-of-concept DAC cycles with ambient air with varying RH (Fig. 4f) and air with 11% RH (Extended Data Fig. 8d), with regeneration by Joule heating in nitrogen. The regenerated PCS-OH was then reused in the next DAC cycle, with a total of four cycles completed. In these experiments, a smaller piece of PCS-OH was used compared to that in Fig. 4b, so that $CO_2$ uptake was limited by the capacity of the sorbent, rather than the quantity of $CO_2$ in the sealed container. The experiments show reversible $CO_2$ capture both in ambient conditions, as well as conditions with controlled RH at 11% (Fig. 3f and Extended Data Fig. 8d). Our results show that our charged-sorbents can be regenerated by direct Joule heating without the need for a secondary conductive support material.

We finally quantified the impact of RH on the $CO_2$ capture performance of PCS-OH, as water deteriorates the $CO_2$ capacity of many OH⁻-based materials[10,27]. Whereas PCS-OH could capture $CO_2$ in humid air (Fig. 4b,f), we found that water co-adsorption is detrimental to the material's $CO_2$ uptake. Under DAC conditions, the $CO_2$ capacity decreased from roughly 0.14 to 0.08 mmol g$^{-1}$ as the RH increased from 11 to 38% (Extended Data Fig. 9a–f). Consistent with this, solid-state $^{13}C$ solid-state NMR experiments at 0.9 bar $CO_2$ and RHs of 53 and 85% showed decreased $CO_2$ chemisorption (0.19 and 0.21 mmol g$^{-1}$, respectively, compared to 0.95 mmol g$^{-1}$ at 0% RH, Extended Data Fig. 9a–f). Joule heating regenerates the $CO_2$ capacity of the sorbent under humid conditions (Fig. 4e,f). Therefore, the reduction in $CO_2$ capacity is probably due to the filling of the hydrophilic pores with $H_2O$ ($H_2O$ uptake isotherm in Extended Data Fig. 9g), blocking access of $CO_2$ to reactive OH⁻ sites[10], rather than due to sorbent degradation.

Using a desorption temperature of 90 °C and the low measured heat capacity of PCS-OH of 0.62 J g$^{-1}$ K$^{-1}$ (Extended Data Fig. 9h), we estimate a minimum electrical energy consumption for sorbent heating of 6.5 and 11.4 GJ ton$^{-1}$ $CO_2$ captured for RH values of 11 and 38%, respectively (equivalent to 1,800 and 3,200 kWh ton$^{-1}$ $CO_2$). These values are comparable to those reported for a range of DAC processes[31]. For example, when using aqueous KOH solutions, the capture of 1 ton of $CO_2$ required either 8.8 GJ natural gas or 5.3 GJ natural gas and 77 kWh electricity in limiting cases[6]. A key advantage of charged-sorbents is that full electrification of the DAC process is possible, thereby avoiding issues with natural gas use and leakage, which can offset a significant fraction of the captured $CO_2$ in traditional DAC processes due to the high global warming potential of methane[32]. Although this may justify the higher operating energy costs for charged-sorbents, future work should be carried out to improve their energy efficiencies. The most straightforward way to achieve this is to increase the sorbent $CO_2$ capacities,

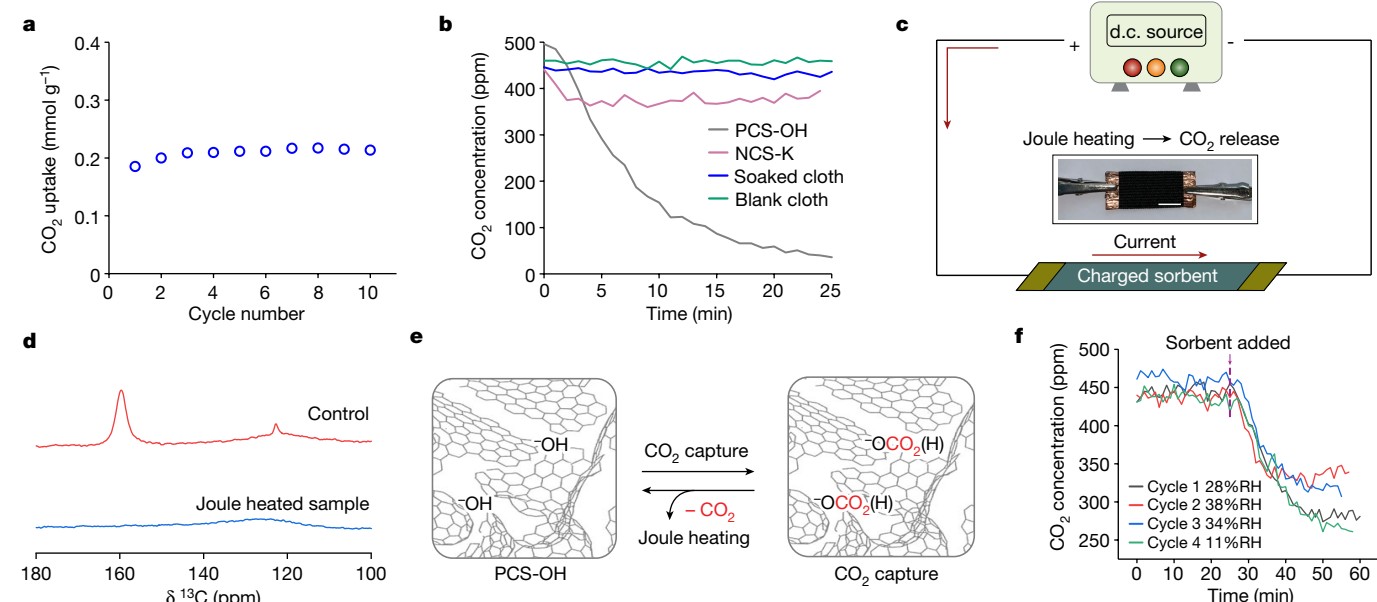

**Fig. 4 | DAC tests and Joule heating regeneration. a,** Cycling capacities of DAC for ten adsorption–desorption cycles for PCS-OH from TGA measurements. Adsorption 30 °C, 60 min, 400 ppm $CO_2$ in dry air; Desorption 130 °C, 60 min, 100% $N_2$. The cycled capacity (difference) is shown. **b,** DAC measurements in a sealed box filled with air at 37% RH. The mass of PCS-OH is 120 mg. **c,** Schematic of Joule heating for $CO_2$ release (scale bar, 0.5 cm, sample dimensions 2 × 1 cm). **d,** $^{13}C$ solid-state NMR spectra of sorbents with $^{13}CO_2$ gas dosing at 0.9 bar: after exposure to the air for 20 min without Joule heating (control sample, red curve) and after 20 min of Joule heating around 90 °C in air (blue curve). MAS rate of 12.5 kHz. **e,** Proposed mechanism for $CO_2$ release from positively (hydroxide) charged-sorbent by means of Joule heating. Schematic adapted from ref. 33, Springer Nature America. **f,** DAC experiments for PCS-OH, with 20 min of Joule heating regeneration under nitrogen between cycles. PCS-OH was added into the box at a time of 25 min (dashed line) for each cycle. The mass of PCS-OH in **f** is 33 mg.

something that we are working towards in our laboratories. With further optimization and scaling, we anticipate that Joule heating regeneration will lead to a rapid temperature–pressure swing DAC process driven by renewable electricity.

## Outlook

Charged-sorbents are a new low-cost materials class with highly tuneable chemical structures. These materials are synthesized through a battery-like charging process that accumulates ions in the sorbent pores, with these species then serving as reactive sites in an adsorption process. Targeting DAC of carbon dioxide as a representative separation, we achieved the electrochemical insertion of hydroxide ions into activated carbon electrodes, with the resulting charged-sorbent material showing enhanced $CO_2$ uptake at low pressures due to chemisorption to form (bi)carbonates. Our materials capture $CO_2$ directly from ambient air and can be regenerated at low temperatures of 90–100 °C. The electrical conductivity of charged-sorbents enables material regeneration by direct Joule heating, with no need for separate heating equipment. The promising oxidative stabilities combined with rapid Joule heating regeneration should enable a DAC process that requires renewable electricity as the only input. Moreover, the low cost of activated carbons and the electrolytes used here is promising from an applications standpoint. Beyond DAC, we anticipate that the readily tuneable nature of our charged-sorbent materials, in which the electrolyte and electrode can easily be varied, will lead to a family of new materials with a wide range of applications.

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

## Methods

### Chemicals

Activated carbon fabric ACC-5092-10 cloth was purchased from Kynol. The cloth was activated for 1 h at 100 °C in a vacuum oven before use. Potassium hydroxide (99%), potassium bicarbonate (99%) and sodium hydroxide (99%) were purchased from Sigma-Aldrich. All chemicals were of analytical grade and directly used as received without further purification.

### Electrochemistry

All electrochemical measurements were performed at room temperature using a BioLogic SP-150 potentiostat and a Biologic BCS-800 Series. A coiled platinum wire (BASi, catalogue no. MW-1033) was used as a counter electrode. In three-electrode measurements, the reference electrode was Hg/HgO (ALS, catalogue no. RE-61AP) with 0.1 M KOH filling solution; the filling solution was exchanged routinely to keep the potential constant. Potentials were converted to the SHE using the correction $E_{SHE} = E_{Hg/HgO} + 0.165$ V.

### Elemental analysis

C, H and N wt% were determined by means of CHN combustion analysis using an Exeter Analytical CE-440, with combustion at 975 °C.

### Volumetric gas sorption measurements

$N_2$ isotherms were collected using an Autosorb iQ gas adsorption analyser at 77 K. The BET surface area was determined by the BET equation and Rouquerol's consistency criteria implemented in AsiQwin. All pore size distribution fittings were conducted in AsiQwin using $N_2$ at 77 K on carbon (slit-shaped pores) quenched solid density functional theory model. $CO_2$ sorption isotherms were also collected on an Autosorb iQ gas adsorption analyser. Isotherms conducted at 25, 35 and 45 °C were measured using a circulating water bath. Samples were activated at 100 °C in vacuum for 15 h before gas sorption measurements.

### Thermogravimetric gas sorption measurements

Thermogravimetric $CO_2$ adsorption experiments were conducted with a flow rate of 60 ml min$^{-1}$ using a TA Instruments TGA Q5000 equipped with a Blending Gas Delivery Module. Samples were activated under flowing $N_2$ for 30 min at various temperatures before cooling to 30 °C and switching the gas stream to $CO_2$ mixtures. Cycling experiments were carried out on a Mettler Toledo TGA/DSC 2 Star$^{ed}$ system equipped with a Huber mini chiller. For tests with high-concentration $CO_2$, the adsorption and desorption of $CO_2$ were performed at 30 and 100 °C for 20 min under 30% $CO_2$ and 70% $N_2$ with a flow rate of 140 ml min$^{-1}$, respectively. For DAC tests, adsorption was carried out at 30 °C for 60 min, with 400 ppm $CO_2$ in dry air, and desorption was carried out at 130 °C for 60 min with 100% $N_2$.

### Adsorption microcalorimetry measurements

The simultaneous measurement of the heat of adsorption and the adsorbed amount of carbon dioxide was performed by means of a heat flow microcalorimeter (Calvet C80 by Setaram), connected to a high-vacuum (residual pressure less than $10^{-4}$ mbar) glass line equipped with a Varian Ceramicell 0–100 mbar gauge and a Leybold Ceramicell 0–1,000 mbar gauge. Before the measurement, both PCS-OH and blank carbon cloth (roughly 150 mg before activation) were activated for 24 h under high vacuum (residual pressure less than $10^{-3}$ mbar) at 100 °C (temperature ramp 3 °C min$^{-1}$). The adsorption microcalorimetry measurements were performed at 30 °C by following a well-established step-by-step procedure described in detail elsewhere[34]. This procedure allowed, during the same experiment, for the determination of both integral heats evolved ($-Q_{int}$) and adsorbed amounts ($n_a$) for small increments of the adsorptive pressure. The partial molar heats obtained for each small dose of gas admitted over the sample were computed

by applying the following ratio: $\Delta Q_{int}/\Delta n_a$, kJ mol$^{-1}$. The (differential) heats of adsorption are then reported as a function of $CO_2$ adsorbed amount, to obtain the (differential) enthalpy changes associated with the proceeding adsorption process. The equilibration time in the microcalorimetric measurement was set to 24 h for small equilibrium pressures (less than 30 mbar), whereas it was reduced to 2 h for larger doses for PCS-OH. The equilibration time was reduced to 2 h (regardless of the equilibrium pressure) for the bare carbon cloth, as equilibration is expected to occur faster in absence of specific adsorption sites.

### X-ray diffraction

Powder X-ray diffraction patterns were collected on a Malvern Panalytical Empyrean instrument equipped with an X'celerator Scientific detector using a non-monochromated Cu Kα source ($\lambda = 1.5406$ Å). The data were collected at room temperature over a $2\theta$ range of 3–80°, with an effective step size of 0.017°.

### Scanning electron microscopy (SEM)

Samples were mounted onto a stainless-steel SEM stub using adhesive carbon tape. SEM imaging was performed on a Tescan MIRA3 FEG-SEM. Analysis of SEM images was conducted using the FIJI ImageJ software.

### NMR spectroscopy

Solid-state NMR experiments were performed with a Bruker Advance spectrometer operating at a magnetic field strength of 9.4 T, corresponding to a $^1$H Larmor frequency of 400.1 MHz. A Bruker 4 mm HX double resonance probe was used in all cases. $^1$H NMR spectra were referenced relative to neat adamantane ($C_{10}H_{16}$) at 1.9 ppm and $^{13}$C NMR spectra were referenced relative to neat adamantane ($C_{10}H_{16}$) at 38.5 ppm (left-hand resonance). All the NMR tests were conducted with a sample magic angle spinning rate of 12.5 kHz. A 90° pulse-acquire sequence was used in each experiment. For $^{13}$C NMR experiments, recycle delays were set to be more than five times the spin-lattice relaxation time for each sample to ensure that the experiments were quantitative.

Charged-sorbents with different water contents were prepared for the NMR characterization. The sorbents were kept in a closed container for 24 h under different RHs. Saturated $Mg(NO_3)_2$ solutions were used to maintain 53% RH at 25 °C, respectively[28].

### Titration measurements

First, 88 mg of sample was immersed in 2 ml of deionized water and sonicated for 20 min at 25 °C. The pH value was then recorded with a pH meter (Insmark IS128C, calibrated with buffer solutions before use) at 25 °C as the initial point. Second, 100 μl of HCl (0.1 M) was slowly added. The mixture was sonicated for 20 min at a constant 25 °C and the pH of the solution was recorded. The second step was repeated until the end of the titration. There was no weight loss due to evaporation during the titration.

### $^{13}CO_2$ dosing for solid-state NMR experiments

Freshly activated samples (75 °C, vacuum oven, 24 h) were packed into 4 mm NMR rotors in air and then evacuated for a minimum of 10 min in a home-built gas manifold[29]. $^{13}$C-enriched $CO_2$ gas (Sigma-Aldrich, less than 3 atom% $^{18}$O, 99.0 atom% $^{13}$C) was then used to dose the samples with gas at room temperature until the gas pressure stabilized, before the rotors were sealed inside the gas manifold with a mechanical plunger.

### DAC tests in sealed chambers

The DAC tests were carried out in a sealed box (volume roughly 600 ml) with a $CO_2$ sensor (Aranet4) to record the concentration of $CO_2$, temperature and RH at every 1 min interval. Before each cycle, the box was exposed to fresh air until the $CO_2$ concentration, RH and temperature stabilized. The sorbent was then placed in the box, which was sealed during measurements.

## Joule heating

After each DAC adsorption step, the sorbent was extracted from the box and connected with an external power source for Joule heating. A BioLogic SP-150 potentiostat was used to vary the electrical input. A constant voltage was applied and adjusted during the experiment to achieve a sample temperature in a range of 85–95 °C under an $N_2$ atmosphere. The temperature was monitored using a thermocouple at a single contact point. After Joule heating regeneration, the electrode was reused for another DAC adsorption cycle.

## Preparation of charged-sorbents

**Three-electrode method.** The charged-sorbents were prepared in a three-electrode configuration with a home-made cell, a Hg/HgO (in 0.1 M KOH) reference electrode and a platinum wire counter electrode. The activated carbon fabric ACC-5092-10 cloth (2 × 2 cm) was charged with a constant potential for 4 h (0.565 V versus SHE for the PCS-OH and −0.235 V versus SHE for NCS-K, respectively) in 40 ml of 6 M KOH(aq.) by means of chronoamperometry (Extended Data Fig. 2). After completing the charging process, the charged cloth was removed and held by plastic tweezers and rinsed with deionized water from a wash bottle for 5 min in total on both sides. Next, 500 ml deionized water in total was used to wash off the residual KOH solution. The rinsed cloth was then placed in the vacuum oven at 75 °C for 24 h to remove the remaining water.

For the soaked control sample, the activated carbon fabric ACC-5092-10 cloth (2 × 2 cm) was soaked in 40 ml of 6 M KOH for 4 h. After soaking, the same rinsing and drying processes used for the charged-sorbents were carried out. As a further control, a dripped cloth sample was prepared by adapting a literature protocol[13]. Here, 200 μl of 6 M KOH was dripped onto the surface of 340.0 mg ACC-10. Subsequently, the dripped samples were left in a Schlenk flask connected to a vacuum to let the samples dry for 72 h at room temperature.

**Two-electrode Swagelok method.** Free-standing carbon films were prepared by adapting the published method in ref. 35. In brief, YP80F activated carbon powder (95 wt%) (Kuraray Chemical) was mixed with polytetrafluoroethylene binder (5 wt%) (Sigma-Aldrich, 60 wt% dispersion in water) in ethanol. The resulting slurry was kneaded and rolled to give a carbon film of roughly 0.25 mm thickness, followed by removing residual solvent at 100 °C in vacuum for at least 24 h. Disc-shaped electrodes were then cut from the carbon films using a 0.25 inch hole punch. Symmetrical Swagelok electrochemical cells were then prepared in Swagelok PFA-820-6 fittings with stainless-steel current collectors, YP80F film electrodes (for both the positive and negative electrodes), 6 M KOH(aq.) electrolyte and a glass fibre separator (Whatman glass microfibre filter (GF/A)). Cells were charged at a constant cell voltage of 0.8 V for 4 h in two-electrode mode, and the positive electrode was then extracted, washed and dried, as above, to yield a charged-sorbent referred to as PCS-OH (YP80F). Three samples from three independent electrochemical cells were combined to provide sufficient material for gas sorption measurements.

## Data availability

All raw experimental data files are available in the Cambridge Research Repository, Apollo, with the identifier https://doi.org/10.17863/CAM.105385.

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

**Acknowledgements** This work was supported by an Engineering and Physical Sciences Research Council New Horizons award (grant no. EP/V048090/1), a Leverhulme Trust Research Project grant (no. RPG-2020-337), a Royal Society Research grant (no. RGS\R2\202125) and a UK Research and Innovation Future Leaders Fellowship to A.C.F. (grant no. MR/T043024/1). Further support came from the US Department of Energy, Office of Science, Office of Basic Energy Sciences under award number DE-SC0021000 and by a Research-to-Impact Fast grant awarded by the 2030 Project and the Cornell Atkinson Center for Sustainability (M.E.Z. and P.J.M.). We acknowledge the support of a Camille Dreyfus Teacher-Scholar Award to PJM (TC-23-048). Support also came from the Program for Guangdong Introducing Innovative and Enterpreneurial Teams (grant no. 2019ZT08L101) and Shenzhen Science and Technology Innovation Commission (grant no. JSGGKQTD20221101115701006). J.W.G. acknowledges the School of the Physical Sciences (Cambridge) for the award of an Oppenheimer Studentship. T.S. received support from the Centre for Climate Repair (Cambridge), X.L. received support from the China Scholarship Council and the Cambridge Trust and T.T. received support from the Development and Promotion of Science and Technology Talents Project, Thailand. M.S and V.C. acknowledge support from the project no. CH4.0 under the Ministero dell'Università e della Ricerca programme 'Dipartimenti di Eccellenza 2023–2027' (grant no. CUP: D13C22003520001). We thank M. Taddei (University of Pisa) for facilitating a new collaboration for this work.

**Author contributions** H.L., H.E., P.J.M. and A.C.F. designed the research. H.L., T.T., X.L., H.E., S.S., T.S., J.T. and C.F. carried out charged-sorbent synthesis and characterization, with J.W.G. providing assistance. H.L., M.E.Z., T.T., X.L., H.E., S.S., T.S., C.F. and S.A.L. carried out gas sorption measurements and analysis, with supervision from P.J.M. and A.C.F. M.S. and V.C. carried out adsorption microcalorimetry measurements and analysis. H.L. and X.L. carried out solid-state NMR spectroscopy measurements and analysis, with supervision from A.C.F. A.C.F. and H.L. wrote the manuscript with contributions from all the authors.

**Competing interests** H.L., H.E. and A.C.F. are authors of a submitted patent on charged-sorbents (PCT/EP2024/055184). The remaining authors declare no competing interests.

**Additional information**
**Correspondence and requests for materials** should be addressed to Alexander C. Forse.

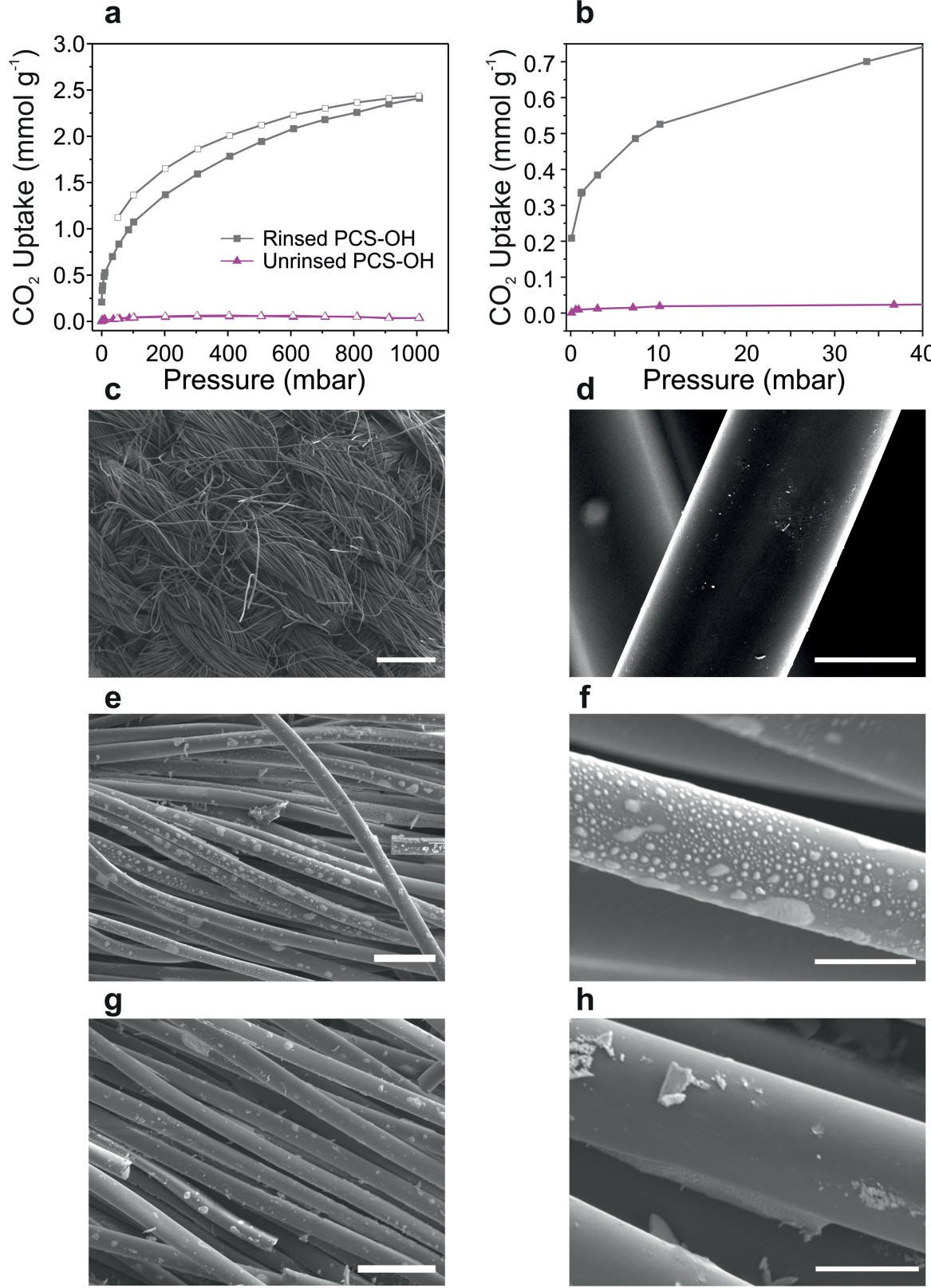

**Extended Data Fig. 1 | The comparison of rinsed and unrinsed PCS-OH charged-sorbents.** (**a**) $CO_2$ adsorption (filled data points) and desorption (hollow data points) isotherms of PCS-OH with the usual water rinsing/washing step, and also without a rinsing step, at 25 °C. (**b**) Low-pressure region of the $CO_2$ adsorption isotherms from (a). Scanning electron microscopic images of activated carbon fabric ACC-5092-10 cloth: (**c**) (**d**) before charging, (**e**) (**f**) unrinsed PCS-OH and (**g**), (**h**) rinsed PCS-OH, with scale bars of (**c**) 500 μm, (**e**) (**g**) 50 μm and (**d**) (**f**) (**h**) 10 μm. The much lower $CO_2$ uptake observed in (a) and (b) when the sample is not rinsed is attributed to the large amount of salt crystals on the carbon surface, which are inactive for $CO_2$ capture.

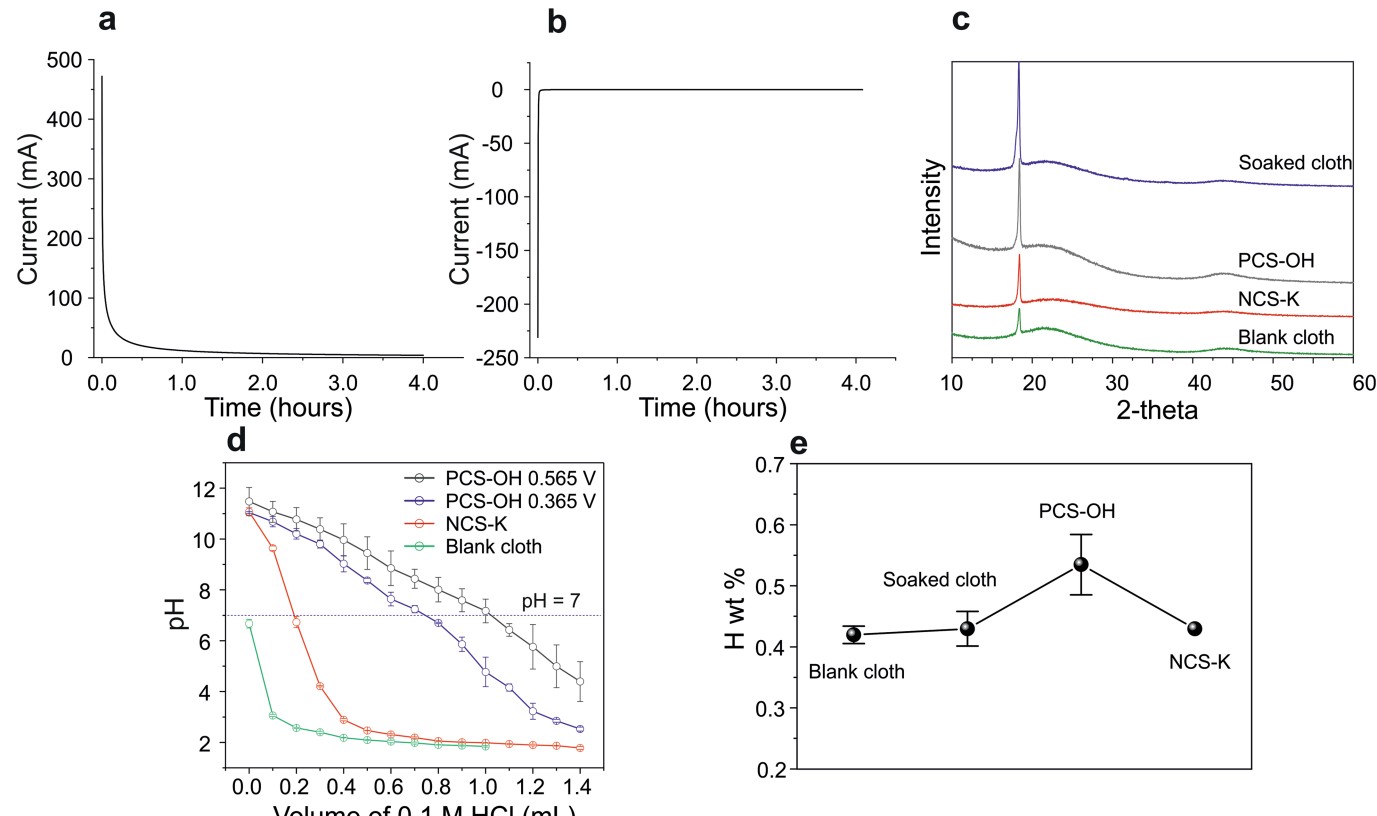

**Extended Data Fig. 2 | The preparation and characterization of charged-sorbents.** The activated carbon fabric ACC-5092-10 cloth was applied with a constant potential (**a**) 0.565 V vs SHE for the PCS-OH. (**b**) −0.235 V vs SHE for the NCS-K. (**c**) Powder X-ray diffraction pattern (Cu Kα radiation, λ = 1.5406 Å) of positively charged-sorbent, negatively charged-sorbent, soaked cloth and blank cloth. The peak at approximately 18° was present in all samples, and its origin is unknown. (**d**) Titration of samples (88 mg pieces) with 0.1 M HCl at 25 °C. PCS-OH 0.565 and PCS-OH 0.365 refer to the samples of PCS-OH prepared at 0.565 V and 0.365 V for 4 h, respectively. Standard deviations are calculated from 2 independent samples. (**e**) The hydrogen weight percentage determined from combustion analysis from blank cloth, soaked cloth, PCS-OH and NCS-K. Standard deviations are calculated from 3 independent samples.

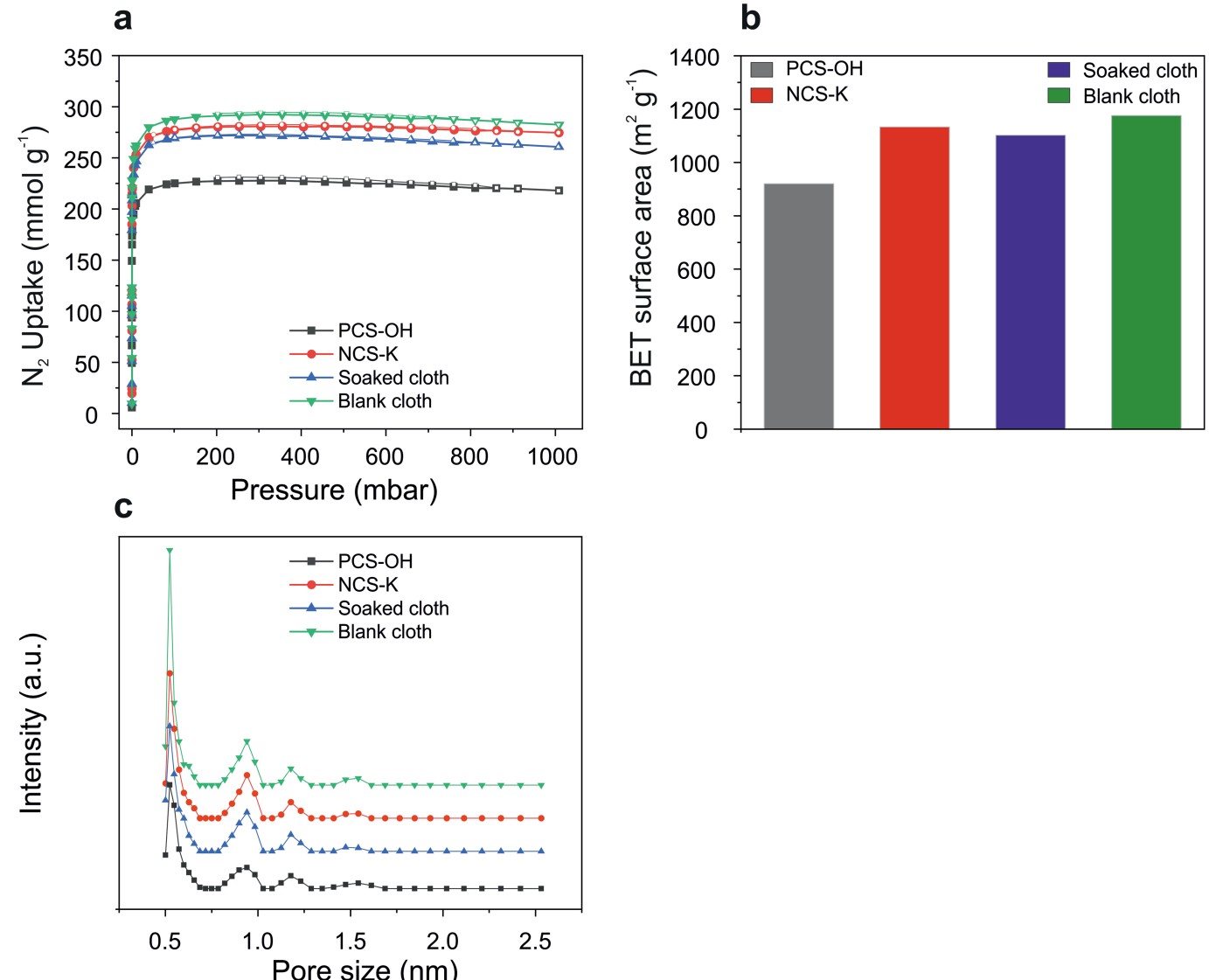

**Extended Data Fig. 3 | Sorbent porosity analysis.** (**a**) $N_2$ adsorption (filled markers) and desorption (open markers) isotherms of PCS-OH, NCS-K, soaked cloth, and blank cloth at 77 K. (**b**) BET surface areas derived from (a). (**c**) Pore size distributions derived from (a) using nonlocal density functional theory (DFT).

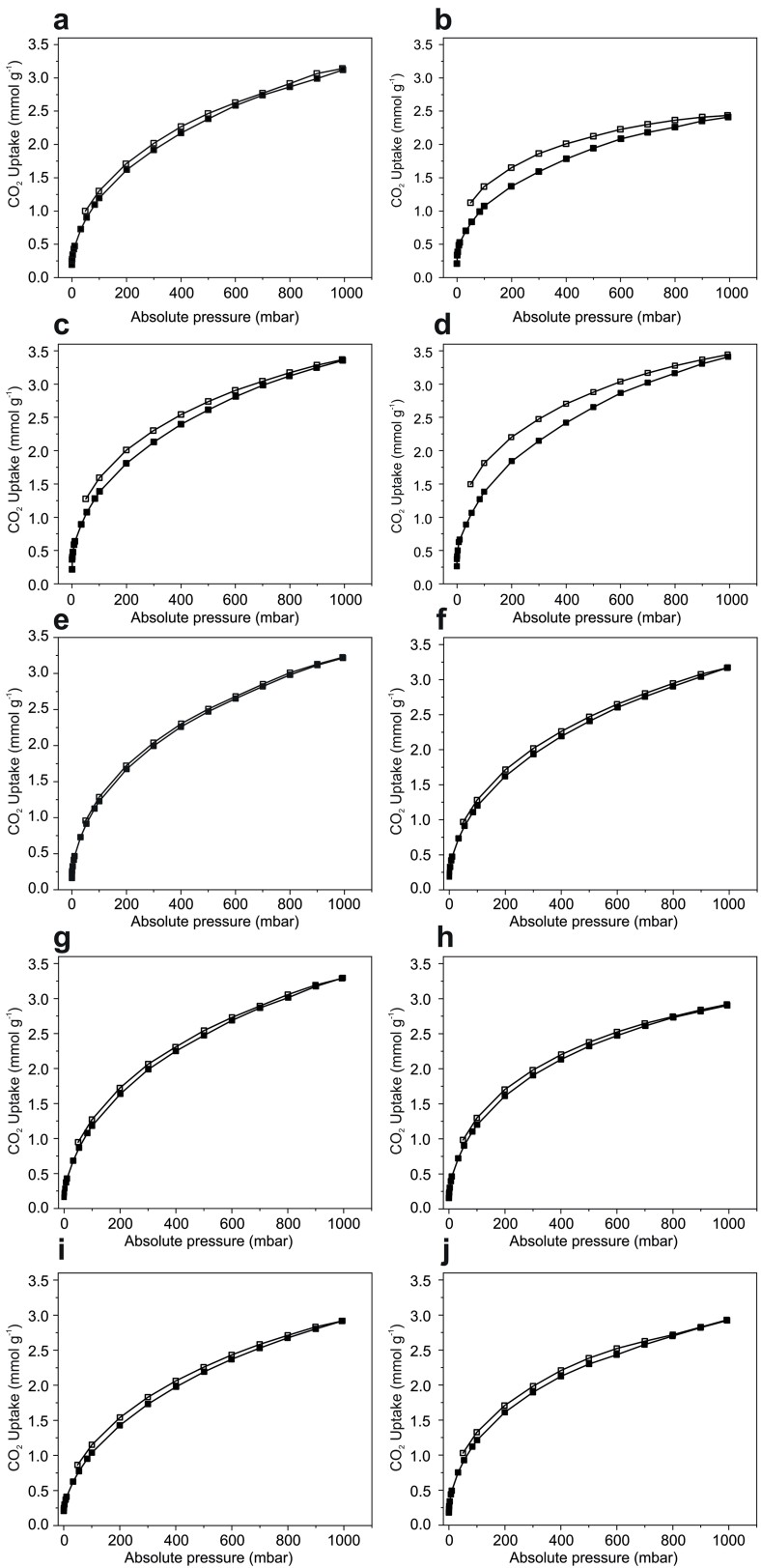

**Extended Data Fig. 4 | CO₂ sorption isotherms of 10 independent PCS-OH samples at 25 °C.** The charging processes were conducted in the same conditions for all the samples. Solid symbols for adsorption curves and hollow symbols for desorption.

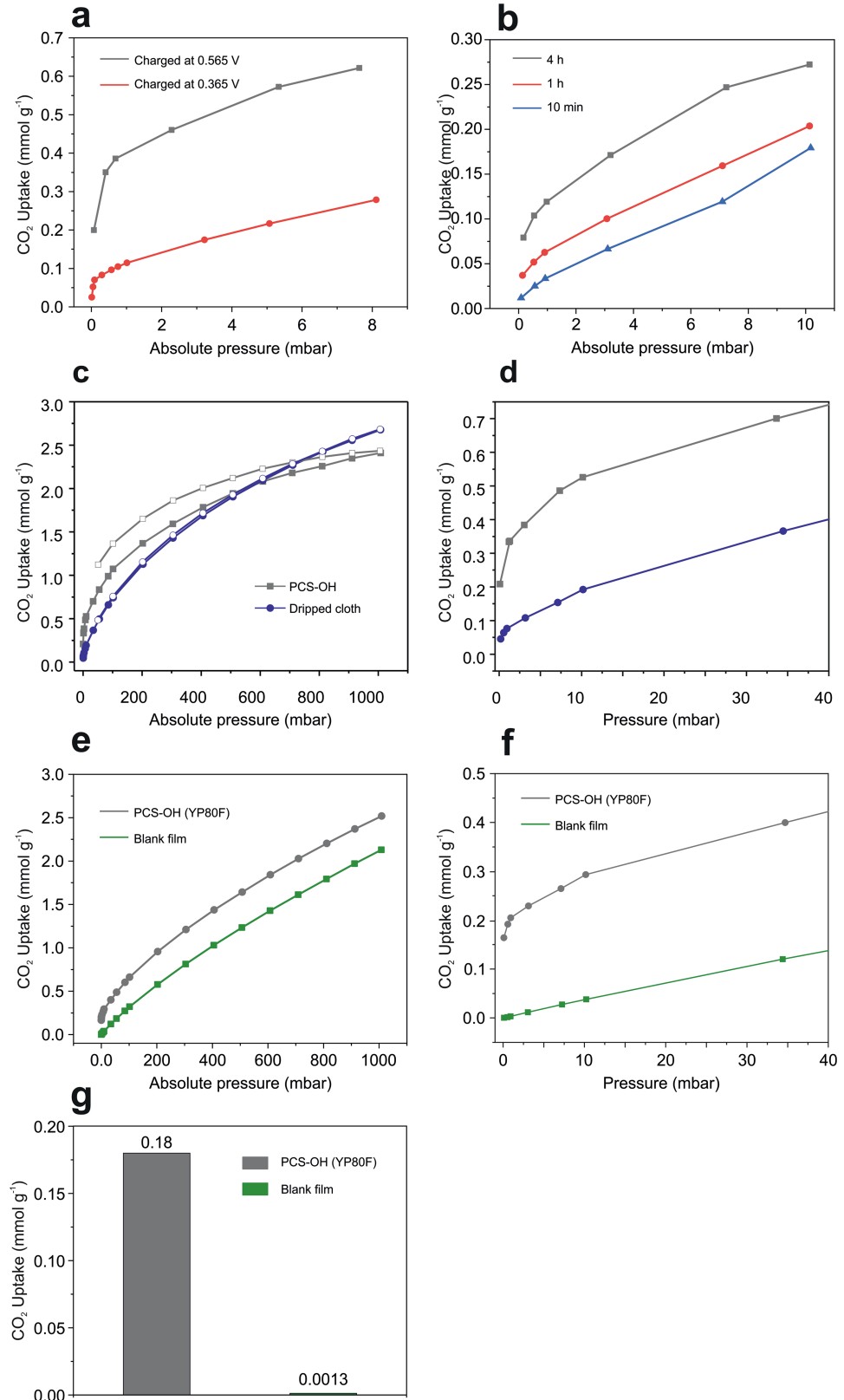

**Extended Data Fig. 5 | CO₂ sorption of PCS-OH sorbents prepared in various conditions.** (**a**) CO₂ uptake (25 °C) of PCS-OH prepared at different potentials V for 4 h in three electrode configuration: 0.565 V and 0.365 V. (**b**) CO₂ uptake (25 °C) of PCS-OH prepared with various charging durations: 4 h, 1 h and 10 min. The charging processes were conducted in a Swagelok cell with potential holds at 0.565 V. (**c**) CO₂ uptake (25 °C) of PCS-OH and a "dripped cloth" sample which was prepared using a method adapted from the literature (see Methods).

(**d**) Low-pressure region of the data in part (c), showing the greater low-pressure CO₂ uptake of PCS-OH at low pressures. (**e**) CO₂ adsorption isotherms of PCS-OH made from YP80F activated carbon film, referred to as PCS-OH (YP80F), and control sample of YP80F activated carbon film at 25 °C. (**f**) Low-pressure region of the CO₂ adsorption isotherms from (e). (**g**) CO₂ uptake of PCS-OH (YP80F) and control sample of YP80F activated carbon at 0.4 mbar and 25 °C.

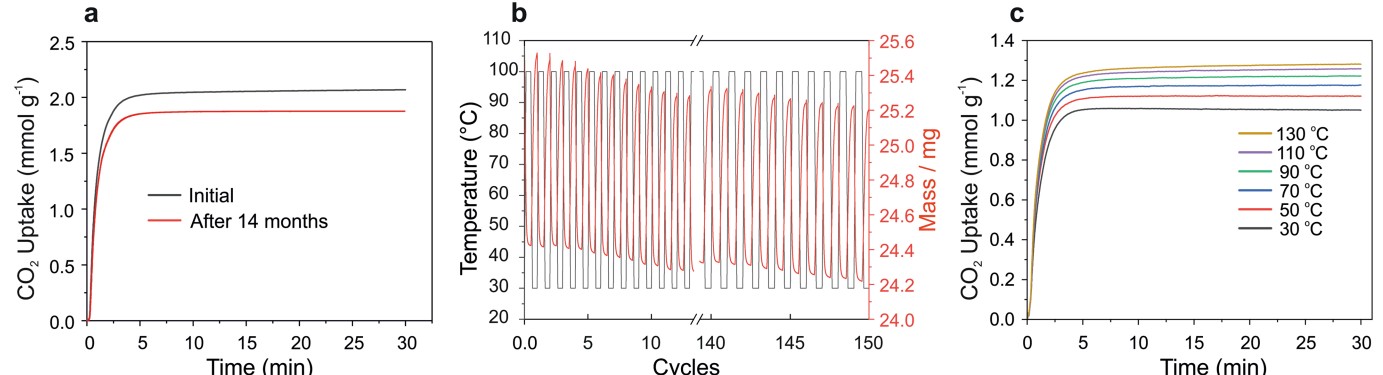

**Extended Data Fig. 6 | The stability of PCS-OH sorbents.** (**a**) Dry, pure $CO_2$ uptake curves at 40 °C and 1 bar $CO_2$ for PCS-OH before and after a period of sample storage in air for 14 months. The loss of capacity after 14 months suggests some degradation of the material. (**b**) Cycling capacities for 150 adsorption/desorption cycles for the PCS-OH in a simulated temperature-pressure swing adsorption process. Adsorption: 30 °C, 20 min, dry 30% $CO_2$ in $N_2$; Desorption: 100 °C, 20 min, dry 30% $CO_2$ in $N_2$. (**c**) $CO_2$ uptake curves at 40 °C and under 50% $CO_2$ in $N_2$ for PCS-OH after activation with various activation temperatures (130 °C, 110 °C, 90 °C, 70 °C, 50 °C and 30 °C).

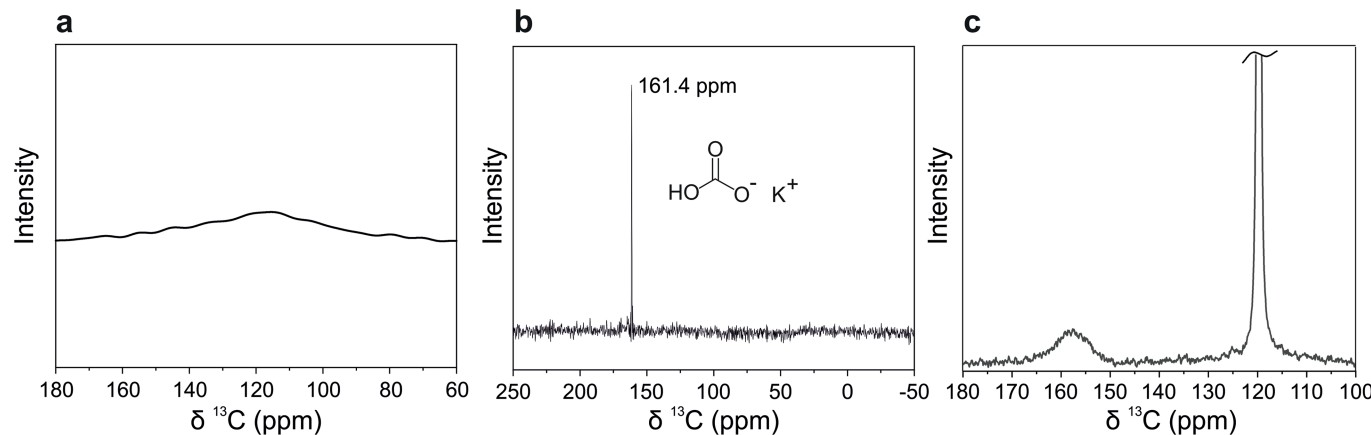

**a** **b** **c**

161.4 ppm

**Extended Data Fig. 7 | $^{13}$C ssNMR study of carbon capture of PCS-OH.**
(**a**) $^{13}$C ssNMR (9.4 T) spectrum of blank activated carbon cloth. A 90° pulse-acquire sequence was applied, and a sample MAS rate of 12.5 kHz was used. A broad resonance arising from sp$^2$-hybridised carbon is observed. (**b**) $^{13}$C ssNMR (9.4 T) spectrum of solid KHCO$_3$. A 90° pulse-acquire sequence was used, with a sample MAS rate of 12.5 kHz. (**c**) $^{13}$C ssNMR (9.4 T) spectrum of PCS-OH prepared with a smaller positive potential of 0.365 V vs. SHE, and loaded with $^{13}$CO$_2$ gas at a pressure of 0.9 bar. A 3.2 mm magic angle spinning HXY probe was used, with a one-pulse $^{13}$C experiment, and the sample spinning rate was 12.5 kHz. The recycle delay was set to be sufficiently long to ensure the spectrum was quantitative. The measurements show a smaller quantity of chemisorbed CO$_2$ compared to NMR measurements on PCS-OH synthesized by applying a potential of 0.565 V vs. SHE (see the main text).

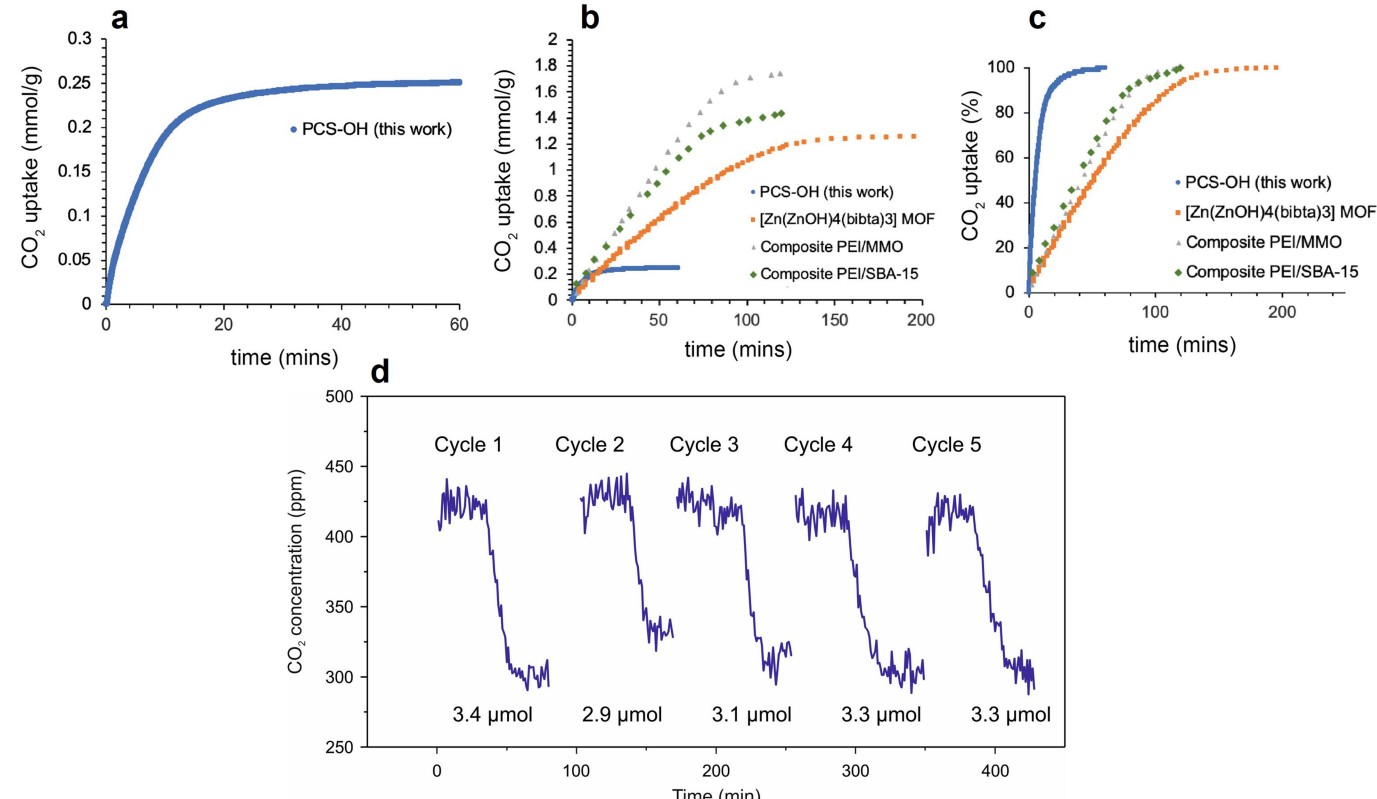

**Extended Data Fig. 8 | Direct air capture uptake kinetics measured by thermogravimetric analysis.** In (**a**), data are shown for PCS-OH, 30 °C, 90 mL/min gas flow, 400 ppm $CO_2$ in air. In (**b**), data are shown for: PCS-OH (14 months after sample preparation, this work), 30 °C, 90 mL/min gas flow, 400 ppm $CO_2$ in air. Zn-$(ZnOH)_4(bibta)_3$ metal-organic framework, 27 °C, 50 mL/min gas flow, 395 ppm of $CO_2$, 21% $O_2$, $N_2$ balance[10]. PEI/MMO composite (PEI67/Mg0.55Al-O), 25 °C, 100 mL/min gas flow, 400 ppm $CO_2$ in $N_2$[36]. PEI/SBA−15 composite (PEI67/SBA−15), 25 °C, 100 mL/min gas flow, 400 ppm $CO_2$ in $N_2$[36]. In (**c**), the data from (**b**) are plotted in terms of percentage uptake for each sorbent. The results in (**b**) show that PCS-OH can capture a comparable quantity of carbon dioxide per unit time to literature sorbents for times of approximately 10 to 15 min. Due to the rapid saturation of PCS-OH shown in (**c**), a relatively short adsorption cycle time should be used for this material. (**d**) DAC cycling experiments for PCS-OH in the sealed chamber, with 20 min Joule heating regeneration under nitrogen between cycles. The RH during DAC tests was controlled by a desiccant (silica gel) at 11% at 25 °C. The mass of PCS-OH used was 30 mg. The $CO_2$ capacities in these tests are listed in the figure, and correspond to gravimetric capacities of 0.11, 0.10, 0.10, 0.11, and 0.11 mmol $g^{-1}$, respectively.

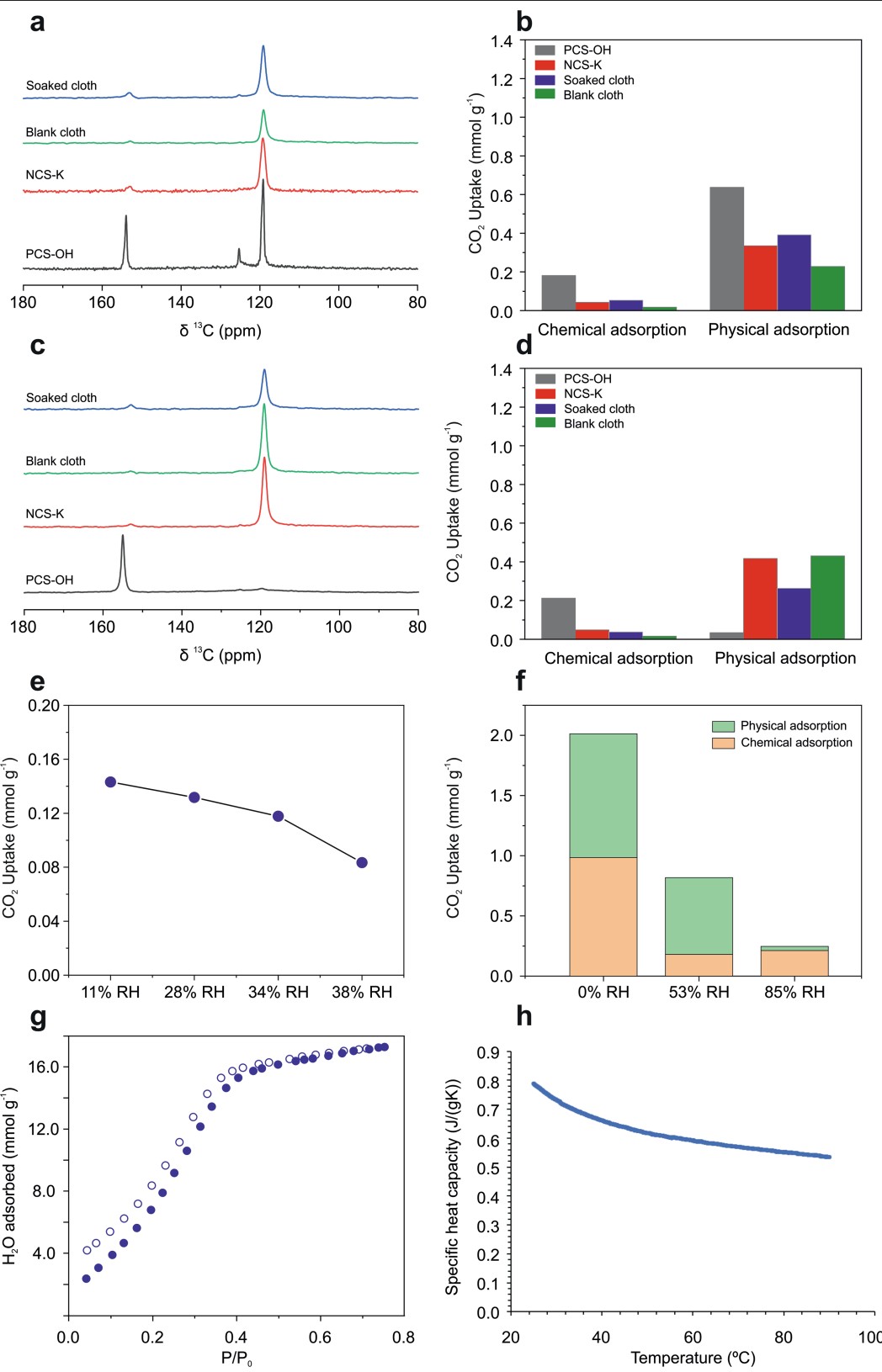

**Extended Data Fig. 9** | See next page for caption.

**Extended Data Fig. 9 | CO$_2$ capture by PCS-OH at various relative humidity conditions.** (**a**) Quantitative $^{13}$C solid-state NMR spectra of sorbents with $^{13}$CO$_2$ gas dosing at 0.9 bar, acquired at a MAS rate of 12.5 kHz. Prior to $^{13}$CO$_2$ dosing, the sorbents were pre-humidified using saturated salt solution (approximately 53% RH at 25 °C). (**b**) CO$_2$ uptake of sorbents via chemical and physical adsorption calculated from resonances at 156 ppm and 119 ppm of (a), respectively. Shown in (**c**) and (**d**) are data analogous to (**a**) and (**b**), except with a relative humidity level of approximately 85% at 25 °C. CO$_2$ uptake of PCS-OH in various relative humidity (RH) at (**e**) direct air capture conditions and (**f**) 0.9 bar CO$_2$. The analysis in part (**e**) is from the sealed-container experiments presented in Fig. 4f of the main text, while the analysis in (**f**) is from solid-state NMR experiments shown in (**a**) and (**c**). (**g**) H$_2$O adsorption (filled circles) and desorption (open circles) isotherms of the PCS-OH at 298.15 K. (**h**) Specific heat capacity measurements from differential scanning calorimetry for PCS-OH, using a DSC2500 from TA instruments. Data is shown for the sample heated from 25 °C to 90 °C after an initial heating and cooling cycle from 5 °C to 200 °C and back to 5 °C. The mass of PCS-OH used was 2.7 mg. The average specific heat capacity between 25 °C and 90 °C is 0.62 J/(gK), and this value was used to estimate the minimum energy required for heating PCS-OH from 25 °C to 90 °C during regeneration, with a value of 40 J/g (see main text).

**Extended Data Table 1 | Comparison of the charge passed for the positive and negative charging processes**

|  | Integrated charge from CA (C) | Charge calculated from capacitance* (C) |
|---|---|---|
| **Positive charging process** | 791.2 | 48 |
| **Negative charging process** | 38.6 | 48 |

During the positive charging process, the integrated charge from chronoamperometry (CA) is larger than that calculated from the capacitance. The charging current remained 12 mA after 4.8 h. This implies that the positive charging process was accompanied by electrochemical carbon oxidation in the KOH electrolyte. However, the porous structure and pore size distribution of the sorbents only changed slightly (around 20% decrease in BET surface area). In contrast, the negative charging process gave the similar charge calculated from both the CA and capacitance measurements, and the charging current decayed to few hundreds nA after 40 min. This finding indicates that a chemical reaction (carbon oxidation) occurred during the positive charging process but not during the negative charging process. * The charge is calculated based on the equation: $Q = C \times U$. Q is the charge passing in the charging process for 1 g carbon fabric ACC-5092-10 cloth, C is the double-layer capacitance (120 F g$^{-1}$ for carbon fabric ACC-5092-10 cloth in the 6 M KOH at 0.4 V), U is the electric potential.