## [Peer Review File · Nature]

Manuscript Title: Capturing Carbon Dioxide from Air with Charged Sorbents

Reviewer Comments & Author Rebuttals

Reviewer Reports on the Initial Version:

Referees' comments:

Referee #1 (Remarks to the Author):

The manuscript by Forse et al. is well written and scientifically appears to be sound. It proposes to use carbon fiber modified to integrate hydroxy moieties for CO₂ capture and specifically direct air capture. These adsorbents were prepared using an electrochemical “charging” method in a highly concentrated KOH media (6M). The difference in CO₂ capture capacity between this modified carbon fiber and its unmodified counterpart under dry conditions is most noticeable at very low CO₂ concentrations, i.e. ~50 mbar and lower. The adsorption capacity at 0.4 mbar CO₂ (400 ppm) remained, however, relatively low at about 0.25 mmol/g compared to other CO₂ capture sorbent such as the ones based on amines. Nevertheless, the stability of the material was good under oxidative conditions at 150°C as would be expected for a material based on OH moieties on a carbon support. However, these materials are of not much use to capture large amounts of CO₂ for sequestration or utilization. There are also problems with humidity. The manuscript is publishable after suitable revision, not in Nature but in your sister journals such as Nature Chemistry or others.

I have a few comments on this manuscript:

On page 5, Figure 1g, please add that dry 30% CO₂ in N₂ was used for both the adsorption and desorption step.

On page 6 line 2, the authors wrote that the adsorption enthalpy was -52 ± 24 kJ/mol. However, looking at Figure S5 e it seems that -52 was more of the maximum value in the figure. How was this value of -52 selected? Concerning Figure S5, this distribution of enthalpy is also seems to differ from similar measurements by other groups. Usually, the value for the enthalpy is the highest at the lowest CO₂ loading, with a progressive decrease when the CO₂ loading increases. This is generally based on the fact that the strongest basic sites (with the highest heat of reaction) react first with CO₂ and as adsorption goes on sites with lower basicity react. Thus, how would the authors explain the shape of this curve?

On page 6 line 8, please add somewhere the name of the material prepared, i.e. PCS-OH (YP80F)

On page 6 line 24. While the stability was remarkable, the affinity towards CO₂ at pressures relevant to DAC was not. Please rephrase. A lot of materials have much more “remarkable” adsorption capacity for DAC.

On page 8 line 6-8, The authors compare the kinetics of their material with the kinetics of some PEI based adsorbents and mention that their material performed 7 times better. However, Measurements for PEI based adsorbents were taken at 25 and 30 °C, whereas the measurements for the carbon material was taken at 40°C. Therefore, it seems logical that the kinetics would be faster at a higher temperature. They should do comparison with the kinetics they obtained at 30 °C to be consistent and

fair (also only 0.1 mmol/g were adsorbed at 40 °C).

In Figure 3c, the scale bar is hardly visible. Please add in the comments the actual size of this sample.

In Figure 3f, why was the mass used only 33 mg? In b it was 120 mg that was used, which seems to be much better for accuracy. These measurements need to be repeated with more adsorbent and the amount of CO₂ quantified. Currently as reported, it is a bit difficult to see the actual difference with the different RH used. Also, why was there no adsorption in the first ~25 minutes in Figure 3f?

On page 9, line 9, the authors wrote that they used dry air. Air containing 11% RH is not dry (value obtained from Figure S14). Please mention the actual RH of the air.

On page 9, line 16 and 17 the authors wrote that the presence of moisture was detrimental to the CO₂ uptake (which is not great for a DAC sorbent). This comment should be substantiated by actual numbers. How much CO₂ was captured at different RH levels?

On line 17 too, the authors need to mention that the ¹³C NMR experiments were conducted with 0.9 bar of CO₂ and not 0.4 mbar because it is not clear.

In Figure S8, please add a figure similar to Fig 1d showing the adsorption for the low-pressure region.

In Figure S9, it is not clear why these adsorption/desorption (30% CO₂ in N₂) conditions were used. They don't seem to correlate to anything else in the text. Please explain.

Referee #2 (Remarks to the Author):

A. The manuscript by Li et al. investigates the use of porous carbon cloth to accumulate hydroxide ions through a battery-like charging process and then chemisorb CO₂ from ambient air and more concentrated workflows. The conductive carbon cloth enables resistive heating, which can rapidly regenerate the sorbent without affecting the stability over several cycles. The authors use a combination of ssNMR spectroscopy, XRD, SEM, TGA, and custom sorption tests to characterize the material and the performance.

B. The key novelty of the work is multifaceted. First, the use of a battery-like charging process to incorporate hydroxide ions into an electrode for the use of CO₂ capture is quite interesting. Certainly, there is much more to explore here (as noted below, can the amount of hydroxide ions be tuned through this process, i.e., increased to raise the chemisorptive capacity?). Next, the use of resistive heating and its effect on the desorption kinetics are very exciting. Again, there is much to be optimized here. Finally, the authors utilize ssNMR to investigate the sorbent material and CO₂ (de)sorption mechanism, all of which are new to the community.

C. The data quality are excellent and the presentation and figure quality are superb. There are some places where the flow could be improved. But, in fact, the figures are very clear and help the text.

D. The use of statistics is appropriate and the authors do a good job of highlighting significant differences. Some of the data do have some noise (e.g., Figures 3b and 3f) and this is not discussed. However, because the authors are looking for deviations that are far from the baseline CO₂ concentration this is not a concern.

E. Conclusions: the authors are a bit modest. Perhaps a useful thought experiment is to compare the costs of heating at 130 °C via an oven for 1 h vs the resistive heating for 1 min. The potential savings

from this approach are tremendous. Of course, a complete TEA would be needed, since the fabrication is not altogether straightforward. The authors could be more aggressive in highlighting the novelty of the work, as noted above.

F. Some suggestions/questions/clarifications are provided here:

- Are hydroxide ions removed with the DI water wash?
- Since part of the decrease in surface area possibly comes from oxidation of the surface, how does this impact the long term stability and/or cyclability of the sorbent?
- How does the uptake capacity compare to the amount of hydroxide ion loaded?
- Why is the capacity lower relative to the blank at high CO₂ partial pressures? Physisorption dominates in this regime and the reduction in surface area limits capacity? Does the capacity reduction follow the change in surface area?
- Why is the cycling done at 30% CO₂ and not DAC (thus far, DAC is the rationale/motivation)?
- Does the oxidative stability depend on the presence of moisture? Are these cycling conditions representative of the true working conditions?
- Is 100 °C achieved by Joule heating? Is this the limit of the heating? Or, is the heating provided by an oven? The regeneration is not well described up to page 6.
- There is no difference in chemical shift between bicarbonate and carbonate? Or, is the chemisorbed CO₂ present in only one form?
- Is the 7.5 mmol/g/h the peak instantaneous rate or the slope of a linearized uptake curve?
- The discussion on page 8 does not discuss the role of RH, but it is mentioned in the figure caption. It is only on page 9 that the discussion of water begins. Perhaps these can be combined somehow?
- Does the water leave the sorbent during regeneration? Have moisture adsorption isotherms and subsequent desorption experiments been performed?

G. The references are appropriate.

H. The authors provide an adequate introduction. It is concise and on point.

Author Rebuttals to Initial Comments:

Referee #1 (Remarks to the Author):

The manuscript by Forse et al. is well written and scientifically appears to be sound. It proposes to use carbon fiber modified to integrate hydroxy moieties for CO₂ capture and specifically direct air capture. These adsorbents were prepared using an electrochemical “charging” method in a highly concentrated KOH media (6M). The difference in CO₂ capture capacity between this modified carbon fiber and its unmodified counterpart under dry conditions is most noticeable at very low CO₂ concentrations, i.e. ~50 mbar and lower.

The adsorption capacity at 0.4 mbar CO₂ (400 ppm) remained, however, relatively low at about 0.25 mmol/g compared to other CO₂ capture sorbent such as the ones based on amines. Nevertheless, the stability of the material was good under oxidative conditions at 150C as would be expected for a material based on OH moieties on a carbon support. However, these materials are of not much use to capture large amounts of CO₂ for sequestration or utilization. There are also problems with humidity. The manuscript is publishable after suitable revision, not in Nature but in your sister journals such as Nature Chemistry or others.

Thank you for the comments. We address the concerns about capturing large amounts of CO₂, and humidity effects, in the comments below (comments 4, 5, and 10).

I have a few comments on this manuscript:

1. On page 5, Figure 1g, please add that dry 30% CO₂ in N₂ was used for both the adsorption and desorption step.

This has been added to the Figure caption:

“Adsorption: 30 °C, 20 min; Desorption: 100 °C, 20 min; dry 30% CO₂ in N₂ was used for both the adsorption and desorption steps.”

2. On page 6 line 2, the authors wrote that the adsorption enthalpy was -52 +/- 24 kJ/mol. However, looking at Figure S5 e it seems that -52 was more of the maximum value in the figure. How was this value of -52 selected? Concerning Figure S5, this distribution of

enthalpy is also seems to differ from similar measurements by other groups. Usually, the value for the enthalpy is the highest at the lowest CO₂ loading, with a progressive decrease when the CO₂ loading increases. This is generally based on the fact that the strongest basic sites (with the highest heat of reaction) react first with CO₂ and as adsorption goes on sites with lower basicity react. Thus, how would the authors explain the shape of this curve?

Thank you for this helpful comment. Having carried out some repeat measurements on the adsorption heats via the variable temperature isotherm and isotherm fitting method, we have concluded that this method does not give reproduceable results for our samples, and we have therefore removed this data.

Therefore, to obtain a more direct and reliable measurement of the adsorption heat as a function of CO₂ loading, we have initiated a new collaboration with researchers at the University of Torino to perform adsorption microcalorimetry measurements. These highly specialised measurements allow the direct measurement of the heat released during gas sorption, rather than relying on isotherm fits as well as fits of a thermodynamic model. The new, more direct, adsorption heat measurements provide physically reasonable results, with a maximal exothermic adsorption heat at low loadings, and a transition to weaker adsorption at higher loadings, as suggested by the reviewer. The new data also corroborate our central hypothesis that the electrochemically inserted hydroxide ions in PCS-OH give rise to enhanced low-pressure CO₂ uptake through chemisorption. We have added the new data to Figure 1, and added a paragraph to discuss the results:

Fig 1f. f) Adsorption microcalorimetry measurements of the differential molar adsorption heats curves related to the adsorption of CO₂ at 30 °C on PCS-OH (grey) and blank cloth (green). The dotted horizontal line represents the standard molar enthalpy of liquefaction of CO₂ at 30 °C: -17 kJ mol⁻¹. Inset: volumetric

isotherms obtained by performing CO₂ adsorption at 30°C on PCS-OH (grey) and blank cloth (green) with the volumetric line coupled to the microcalorimeter.

From the revised text:

“To investigate the nature of CO₂ sorption in **PCS-OH** we performed microcalorimetry tests which allow the direct quantification of the heat released during CO₂ uptake measurements (**Fig. 1f**). For the blank carbon cloth control, the measured CO₂ adsorption heat is between -28 and -20 kJ mol⁻¹, consistent with CO₂ physisorption.²⁰ Excitingly, a large increase in the adsorption heat is observed for **PCS-OH** relative to the blank carbon, consistent with CO₂ chemisorption. The measured adsorption heats gradually decrease from a value of -137 kJ mol⁻¹ at zero coverage (extrapolated value) to -33 kJ mol⁻¹ at a coverage of 0.8 mmol g⁻¹ suggestive of bicarbonate formation at a distribution of hydroxide sites. Importantly, the adsorption heats compare well with previous reports for bicarbonate formation in porous materials and metal oxides,^{10,11,21} lending support that the electrochemically inserted hydroxides in **PCS-OH** serve as CO₂ chemisorption sites to enhance low-pressure uptake.”

We have also revised the experimental methods section to provide the details of the adsorption microcalorimetry measurements (not shown here).

3. On page 6 line 8, please add somewhere the name of the material prepared, i.e. PCS-OH (YP80F)

This has been added. From the revised text:

“As a second example, powdered YP80F activated carbon was first prepared into a free-standing electrode film before synthesising a charged-sorbent referred to as **PCS-OH (YP80F)** (Fig. S10, see Methods **Section 2**).”

4. On page 6 line 24. While the stability was remarkable, the affinity towards CO₂ at pressures relevant to DAC was not. Please rephrase. A lot of materials have much more “remarkable” adsorption capacity for DAC.

This sentence has been rephrased:

*“Overall, these data support that **PCS-OH** exhibits remarkable stability and promising CO₂ uptake at pressures relevant to DAC.”*

We have also carried out new experiments to compare the CO₂ uptake of charged-sorbents to work on activated carbons impregnated with aqueous KOH solutions (Figure 1e, Figure S1a,b). The results further highlight that charged-sorbents have best-in-class in CO₂ capacities at 0.4 mbar among (modified) activated carbons, supporting their potential as DAC sorbents. From the revised text:

“PCS-OH also showed enhanced low-pressure CO₂ uptake compared to a “dripped cloth” sample, which was preparing by adapting a literature approach for impregnating activated carbon with metal hydroxides solutions (Fig. 1e, Fig. S9, Methods).”¹³”

The new data has been added to Figure S9:

Fig. S9 (a) CO₂ uptake (25 °C) of PCS-OH and a “dripped cloth” sample which was prepared using a method adapted from the literature (see Methods). **(b)** Low-pressure region of the data in part (a), showing the greater low-pressure CO₂ uptake of PCS-OH at low pressures.

Finally, we would like to emphasise that while there are literature sorbents that have higher DAC capacities than PCS-OH, the rapid kinetics for PCS-OH should enable competitive productivities (i.e. quantities of CO₂ captured per unit time per amount of sorbent). From the revised text:

“The DAC kinetics for PCS-OH are also excellent, with a faster CO₂ capture rate than several benchmark sorbents (Fig. S17). While other sorbents have been reported with higher DAC capacities, the rapid kinetics for PCS-OH should enable a highly productive DAC process.”

And from the revised conclusion:

“The excellent CO₂ uptake kinetics combined with rapid Joule heating regeneration should enable a highly productive energy-efficient DAC process that requires renewable electricity as the only input.”

In addition, with additional experiments we have found that the CO₂ capacity is correlated with the amount of OH⁻ in the sorbent (see reviewer 2, comment iii). We therefore envisage that it will be possible to increase the OH⁻ loading in the PCS-OH to further improve the adsorption capacity for DAC in the future.

5. On page 8 line 6-8, The authors compare the kinetics of their material with the kinetics of some PEI based adsorbents and mention that their material performed 7 times better. However, Measurements for PEI based adsorbents were taken at 25 and 30 °C, whereas the measurements for the carbon material was taken at 40 °C. Therefore, it seems logical that the kinetics would be faster at a higher temperature. They should do comparison with the kinetics they obtained at 30 °C to be consistent and fair (also only 0.1 mmol/g were adsorbed at 40 °C).

At the reviewers’ suggestion, we have carried out new kinetics measurements at 30 °C, and obtained a capacity that is more consistent with our isotherm measurements.

We have carried out a more complete comparison of the kinetics for PCS-OH with literature samples by extracting raw data from previous publications where possible.

We have added the new kinetics data and the comparison plot in the SI. These results highlight the promising kinetic performance of PCS-OH:

Fig. S17. Direct air capture uptake kinetics measured by thermogravimetric analysis. In (a), data are shown for PCS-OH, 30 °C, 90 mL/min gas flow, 400 ppm CO₂ in air. In (b), data are shown for: PCS-OH (14 months after sample preparation, this work), 30 °C, 90 mL/min gas flow, 400 ppm CO₂ in air. Zn-(ZnOH)₄(bibta)₃ metal-organic framework, 27 °C, 50 mL/min gas flow, 395 ppm of CO₂, 21% O₂, N₂ balance.¹ PEI/MMO composite (PEI67/Mg0.55Al-O), 25 °C, 100 mL/min gas flow, 400 ppm CO₂ in N₂.² PEI/SBA-15 composite (PEI67/SBA-15), 25 °C, 100 mL/min gas flow, 400 ppm CO₂ in N₂.² These results highlight the excellent kinetic performance of PCS-OH.

We therefore believe that these materials could indeed capture significant quantities of CO₂ for sequestration and utilization. While their CO₂ uptake capacities are not as large as some other DAC sorbents (e.g. MOFs, amine polymers etc.), their rapid kinetics, and their rapid regeneration by Joule heating should enable large amounts of CO₂ to be accumulated. From the revised text:

“The excellent CO₂ uptake kinetics combined with fast Joule heating regeneration should enable a rapid energy-efficient DAC process that requires renewable electricity as the only input.”

6. In Figure 3c, the scale bar is hardly visible. Please add in the comments the actual size of this sample.

The dimensions of the sample have been added to the figure caption. From the revised text:

“...Scale bar: 0.5 cm, sample dimensions: 2 × 1 cm...”

7. In Figure 3f, why was the mass used only 33 mg? In b it was 120 mg that was used, which seems to be much better for accuracy. These measurements need to be repeated with more adsorbent and the amount of CO₂ quantified. Currently as reported, it is a bit difficult to see the actual difference with the different RH used.

A lower sample mass was intentionally selected for the experiments in Figure 3f so that the CO₂ uptake would be limited by the capacity of the sorbent, rather than the quantity of carbon dioxide in the sealed chamber. Under these conditions, the measurements enable the calculation of CO₂ capacities, which are now included in Fig. S19, and are discussed below in response to point 10. These conditions also ensure that the sorbent is saturated with CO₂ prior to the Joule heating regeneration step. If a large piece of sorbent was used, the sorbent may not be saturated with CO₂, and it would be unclear whether or not regeneration was achieved with Joule heating.

To explain this point, the following text has been added to the manuscript:

*“In these experiments a smaller piece of PCS-OH was used compared to that in **Fig. 3b**, so that CO₂ uptake was limited by the capacity of the sorbent, rather than the quantity of CO₂ in the sealed container.”*

8. Also, why was there no adsorption in the first ~25 minutes in Figure 3f?

The sorbent was added to the sealed container after 25 minutes, so it is only at this point that adsorption is expected. Apologies that this was unclear in our submitted manuscript.

This point has now been clarified by adding a dashed line and the text “sorbent added” to Figure 3f. From the revised Figure 3:

Text was also added to the figure caption to clarify this point:

“PCS-OH was added into the box at a time of 25 min (dashed line) for each cycle.”

9. On page 9, line 9, the authors wrote that they used dry air. Air containing 11% RH is not dry (value obtained from Figure S14). Please mention the actual RH of the air.

Thank you for spotting this error. This has been addressed in the revised text:

*“We then carried out proof-of-concept DAC cycles with ambient air with varying RH (**Fig. 3f**), and air with 11% RH (Fig. S18), with regeneration by Joule heating in nitrogen.”*

“The experiments display reversible CO₂ capture both in ambient conditions, as well as conditions with controlled RH at 11% (Fig 3f, Fig. S18).”

10. On page 9, line 16 and 17 the authors wrote that the presence of moisture was detrimental to the CO₂ uptake (which is not great for a DAC sorbent). This comment should be substantiated by actual numbers. How much CO₂ was captured at different RH levels?

Additional analysis has been added to the manuscript on this in the Supplementary Information, and the following text has been added to the manuscript:

“We finally quantified the impact of relative humidity (RH) on the CO₂ capture performance of PCS-OH... Under DAC conditions, the CO₂ capacity decreased from approximately 0.14 to 0.08 mmol/g as the RH increased from 11 to 38% (Fig. S19). Consistent with this, solid-state ¹³C solid-state NMR experiments at 0.9 bar CO₂ and RHs of 53% and 85% showed decreased chemisorption (0.19 and 0.21 mmol/g respectively, compared to 0.95 mmol/g at 0% RH, Fig. S19).”

New measurements and new analysis were added in the supplementary information (Fig. S19, c, d, e, f):

Fig. S19. (a) Quantitative ^{13}C solid-state NMR spectra of sorbents with $^{13}\text{CO}_2$ gas dosing at 0.9 bar, acquired at a MAS rate of 12.5 kHz. Prior to $^{13}\text{CO}_2$ dosing, the sorbents were pre-humidified using saturated salt solution (approximately 53% RH at 25 °C). (b) CO_2 uptake of sorbents via chemical and physical adsorption calculated from resonances at 156 ppm and 119 ppm of (a), respectively. Shown in (c) and (d) are data analogous to (a) and (b), except with a relative humidity level of approximately 85% at 25 °C. CO_2 uptake of PCS-OH in various relative humidity (RH) at (e) direct air capture conditions and (f) 0.9 bar CO_2 . The analysis in part (e) is from the sealed-container experiments presented in Fig. 3f of the main text, while the analysis in (f) is from solid-state NMR experiments shown in (a) and (c).

Importantly, our new materials still perform DAC under humid conditions, albeit at reduced CO_2 capacity. An active area of ongoing research in our group is on improving DAC performance under humid conditions, for example by surface modification of the carbon to make it more hydrophobic. We are also exploring these materials for moisture-swing regeneration, which could provide another means to achieve an efficient direct air capture process.

11. On line 17 too, the authors need to mention that the ^{13}C NMR experiments were conducted with 0.9 bar of CO_2 and not 0.4 mbar because it is not clear.

This has been clarified. From the revised text:

“Consistent with this, solid-state ^{13}C solid-state NMR experiments at 0.9 bar CO_2 and RHs of 53% and 85% showed decreased chemisorption (0.19 and 0.21 mmol/g respectively, compared to 0.95 mmol/g at 0% RH, Fig. S19).”

We have added similar clarification to page 7, on the NMR experiments under dry conditions:

“To investigate the mechanistic pathway responsible for strong CO_2 binding in PCS-OH, we collected ^{13}C solid-state NMR (ssNMR) spectra for PCS-OH and the control samples after dosing with $^{13}\text{CO}_2$ gas at 0.9 bar (Fig. 2a).”

12. In Figure S8, please add a figure similar to Fig 1d showing the adsorption for the low-pressure region.

A new figure panel (part b) has been added to this Figure, showing the low-pressure region (the new figure is Fig. S10). This better supports our point from the main text, that a second example charged-sorbent has been prepared, which also shows significantly enhanced low-pressure CO₂ uptake compared to the as purchased carbon material.

From the revised SI:

“

Fig S10 (a) CO₂ adsorption isotherms of PCS-OH made from YP80F activated carbon film, referred to as PCS-OH (YP80F), and control sample of YP80F activated carbon film at 25 °C. (b) Low-pressure region of the CO₂ adsorption isotherms from (a). (c) CO₂ uptake of PCS-OH (YP80F) and control sample of YP80F activated carbon at 0.4 mbar and 25 °C.”

13. In Figure S9, it is not clear why these adsorption/desorption (30% CO₂ in N₂) conditions were used. They don't seem to correlate to anything else in the text. Please explain.

Figure S9 (the new figure is Fig. 12) is the raw data corresponding to Figure 1g in the main text. This is addressed in the below revised text:

*“As a further stability assay prior to DAC tests, **PCS-OH** was subjected to 150 adsorption and desorption cycles using TGA under concentrated CO₂ conditions (30% CO₂ in N₂) (Fig. 1g and Fig. S12).”*

Referee #2 (Remarks to the Author):

A. The manuscript by Li et al. investigates the use of porous carbon cloth to accumulate hydroxide ions through a battery-like charging process and then chemisorb CO₂ from ambient air and more concentrated workflows. The conductive carbon cloth enables resistive heating, which can rapidly regenerate the sorbent without affecting the stability over several cycles. The authors use a combination of ssNMR spectroscopy, XRD, SEM, TGA, and custom sorption tests to characterize the material and the performance.

B. The key novelty of the work is multifaceted. First, the use of a battery-like charging process to incorporate hydroxide ions into an electrode for the use of CO₂ capture is quite interesting. Certainly, there is much more to explore here (as noted below, can the amount of hydroxide ions be tuned through this process, i.e., increased to raise the chemisorptive capacity?). Next, the use of resistive heating and its effect on the desorption kinetics are very exciting. Again, there is much to be optimized here. Finally, the authors utilize ssNMR to investigate the sorbent material and CO₂ (de)sorption mechanism, all of which are new to the community.

C. The data quality are excellent and the presentation and figure quality are superb. There are some places where the flow could be improved. But, in fact, the figures are very clear and help the text.

D. The use of statistics is appropriate and the authors do a good job of highlighting significant differences. Some of the data do have some noise (e.g., Figures 3b and 3f) and this is not discussed. However, because the authors are looking for deviations that are far from the baseline CO₂ concentration this is not a concern.

E. Conclusions: the authors are a bit modest. Perhaps a useful thought experiment is to compare the costs of heating at 130 °C via an oven for 1 h vs the resistive heating for 1 min. The potential savings from this approach are tremendous. Of course, a complete TEA would be needed, since the fabrication is not altogether straightforward. The authors could be more aggressive in highlighting the novelty of the work, as noted above.

Thanks for the supportive comments. We believe a key strength of our charged-sorbent materials lies in their rapid CO₂ uptake kinetics and fast regeneration using Joule heating. We have added an additional sentence to highlight this point in the conclusions:

“The excellent CO₂ uptake kinetics combined with rapid Joule heating regeneration should enable a highly productive energy-efficient DAC process that requires renewable electricity as the only input.”

We have also emphasised the parallel between charged-sorbent synthesis and the charging of a battery in the conclusion, to further highlight the novelty of our work. We believe this point will appeal to a general audience and can inspire other research in this new area:

“These materials are synthesized through a battery-like charging process that accumulates ions in the sorbent pores, with these species then serving as reactive sites in an adsorption process.”

We agree with the reviewer that the comparison of Joule heating regeneration and more conventional convective heating would be very interesting, though we prefer to wait to scale-up our materials and process first, so we can make a more meaningful and fair comparison.

F. Some suggestions/questions/clarifications are provided here:

(i) • Are hydroxide ions removed with the DI water wash?

Our understanding is that the DI water wash does not remove hydroxide ions from the carbon pores, as these are strongly held by the positive charges on the carbon surface. However, the DI water wash is essential to remove excess KOH solution after the electrode is removed from the electrochemical cell. Washing limits the formation of salt crystals on the carbon cloth surface, and improves CO₂ capture performance. We have added the following sentence to the manuscript to clarify this point:

“Following charging, the electrode was removed from the cell, and was washed to remove excess KOH and minimize salt formation on the surface of the carbon cloth (Fig. S1).”

We have recorded new adsorption isotherm measurements and have added these to the SI, to show the detrimental impact of omitting the DI water wash. We have also added additional SEM images, which provide evidence that washing removes salt crystals from the carbon surface.

From Figure S1: “

Fig. S1. a) CO₂ adsorption (filled data points) and desorption (hollow data points) isotherms of PCS-OH with the usual water rinsing/washing step, and also without a rinsing step, at 25 °C. b) Low-pressure region of the CO₂ adsorption isotherms from (a). Scanning electron microscopic images of activated carbon fabric ACC-5092-10 cloth: (c) (d) before charging, (e) (f) unrinsed PCS-OH, and (g) (h) rinsed PCS-OH, with scale bars of (c) 500 μm, (e) (g) 50 μm and (d) (f) (h) 10 μm. The much lower CO₂ uptake observed in (a) and (b) when the sample is not rinsed is attributed to the large amount of salt crystals on the carbon surface, which are inactive for CO₂ capture.

(ii) • Since part of the decrease in surface area possibly comes from oxidation of the surface, how does this impact the long term stability and/or cyclability of the sorbent?

Our cyclability tests (**Fig. 1g**) showed good cycling stability over 150 cycles.

We have also now additional experiments to test the long-term stability of PCS-OH. A sample which was stored in air for 14 months was tested with a TGA CO₂ uptake assay, which reveals some loss of capacity over time. From the revised manuscript:

“We do however note that 10% of its capacity was lost after sample storage for 14 months (**Fig. S11**).”

From the revised SI:

“

Fig. S11. Dry, pure CO₂ uptake curves at 40 °C and 1 bar CO₂ for PCS-OH before and after a period of sample storage in air for 14 months. The loss of capacity after 14 months suggests some degradation of the material.”

However, in spite of this, PCS-OH still showed good DAC performance after the 14-month sample aging, underscoring its promising stability for this application. From the revised text:

“Importantly, **PCS-OH** can still do DAC (Fig. S17a) at comparable capacity after 14 months, despite the small capacity lost seen under pure CO₂ conditions.”

(iii) • How does the uptake capacity compare to the amount of hydroxide ion loaded?

We have now done additional experiments that provide evidence that the low-pressure CO₂ uptake is positively correlated with the amount of hydroxide ions loaded in the material.

First, titration measurements have been performed and added to the SI as a probe of hydroxide ion content in PCS-OH materials. From the revised text:

“Further support for the accumulation of OH⁻ species in PCS-OH was provided by titration experiments, where 1.2 mmol/g of HCl was required to neutralise PCS-OH, compared to 0.2 mmol/g for NCS-K (Fig. S4).”

From the revised SI: “

Fig. S4. Titration of samples (88 mg pieces) with 0.1 M HCl at 25 °C. PCS-OH 0.565 and PCS-OH 0.365 refer to the samples of PCS-OH prepared at 0.565 V and 0.365 V for 4 h, respectively. Standard deviations are calculated from 2 independent samples.”

and from the main text:

*“These results further allow us to estimate a lower limit for the hydroxide content in **PCS-OH** of 0.95 mmol g⁻¹ (assuming a 1:1 reactivity of CO₂ and hydroxide), which is comparable to the value of 1.2 mmol g⁻¹ from titration (Fig. S4).”*

Second, experiments on samples synthesized using less positive charging potentials gave rise to lower CO₂ uptake capacities, and lower hydroxide ion content. The following sentence has been added to the manuscript to make this point:

“Experiments on samples synthesized using less positive charging potentials and shorter charging times gave smaller CO₂ uptake at low pressures, due to the incorporation of fewer hydroxide ions (Figs. S4, 8).”

Third, we have carried out an additional NMR spectroscopy experiment to determine the reactive hydroxide content in a sample charged at potential of 0.365 V vs. SHE (which is a less positive potential than our optimised value of 0.565 V). The NMR experiments support a lower amount of loaded hydroxide in this case, as anticipated, while the CO₂ adsorption isotherms show lower capacities at low pressure. From the revised text:

“Similar NMR spectroscopy experiments on PCS-OH prepared at a lower charging potential (0.365 V vs. SHE, instead of our optimised value of 0.565 V vs. SHE) provided a much lower limiting hydroxide content of 0.38 mmol g⁻¹, and an estimated molecular formula of (OH)C₂₁₈ (Fig. S16). The lower hydroxide content of samples prepared in this way led to lower CO₂ uptakes at low pressures (Fig. S8a), supporting a positive correlation between hydroxide content and low-pressure CO₂ uptake.”

And the new NMR data has been added to the Supplementary Information:

“

Fig. S16. ^{13}C ssNMR (9.4 T) spectrum of PCS-OH prepared with a smaller positive potential of 0.365 V vs. SHE, and loaded with $^{13}\text{CO}_2$ gas at a pressure of 0.9 bar. A 3.2 mm magic angle spinning HXY probe was used, with a one-pulse ^{13}C experiment, and the sample spinning rate was 12.5 kHz. The recycle delay was set to be sufficiently long to ensure the spectrum was quantitative. The measurements show a smaller quantity of chemisorbed CO_2 compared to NMR measurements on PCS-OH synthesized by applying a potential of 0.565 V vs. SHE (see the main text).”

In the future, we hope to find ways to increase the hydroxide content in charged-sorbents, as a means to improve their capacities for direct air capture.

(iv) • Why is the capacity lower relative to the blank at high CO_2 partial pressures?
 Physisorption dominates in this regime and the reduction in surface area limits capacity?
 Does the capacity reduction follow the change in surface area?

We believe there are two possible reasons for the decrease in capacity at high pressures:

- (i) As the reviewer suggests, the lower surface area of PCS-OH relative to the blank film most likely contributes to the capacity reduction.
- (ii) The PCS-OH samples has additional mass due to the added hydroxide groups, though this is a minor effect ($\sim 2\%$, assuming a molecular formula of $(\text{OH})\text{C}_{86}$).

After some consideration, we decided not to comment on this point in the manuscript. This is because our focus is on the low-pressure uptake, and we also cannot unambiguously separate chemisorption and physisorption processes from the isotherms alone, meaning any added text would be somewhat speculative here.

(v) • Why is the cycling done at 30% CO₂ and not DAC (thus far, DAC is the rationale/motivation)?

The initial cycling at 30% CO₂ was performed as a convenient assay of the material stability. We have modified the text to make this more clear:

“As a further stability assay prior to DAC tests, PCS-OH was subjected to 150 adsorption and desorption cycles using TGA under concentrated CO₂ conditions (30% CO₂ in N₂) (Fig. 1g and Fig. S12).”

We note that we have also performed DAC cycling tests in **Fig. 3a**, which show the stable cycling of the material.

(vi) • Does the oxidative stability depend on the presence of moisture? Are these cycling conditions representative of the true working conditions?

Our DAC tests in Fig. 3f and Fig. S18 are in the presence of both oxygen and water, and show stable cycling over five cycles. Longer term cycling experiments in the presence of both water and oxygen should be carried out in the future, and we are buying new equipment to enable these measurements.

(vii) • Is 100 °C achieved by Joule heating? Is this the limit of the heating? Or, is the heating provided by an oven? The regeneration is not well described up to page 6.

The initial gas sorption experiments use convention heating in a furnace, and joule heating is only introduced in the section titled “Demonstration of DAC and regeneration by Joule heating”. To clarify this, the following sentence was modified in the manuscript:

“In addition to the promising gas sorption results, thermogravimetric analysis (TGA) measurements (using a heated furnace) indicate that PCS-OH has good thermal and oxidative stability.”

(viii) • There is no difference in chemical shift between bicarbonate and carbonate? Or, is the chemisorbed CO₂ present in only one form?

With ¹³C NMR is it generally not possible to conclusively differentiate between bicarbonate and carbonate, due to their rapid chemical exchange on the NMR timescale. We have added a reference on this, and have added the below text to clarify this point:

“Note that carbonate and bicarbonate species are often in fast exchange on the NMR timescale, so we cannot readily discriminate between these species.²⁹ Regardless, observation of this chemisorption resonance provides strong evidence that the hydroxide sites incorporated through our electrochemical synthesis chemically react with CO₂.”

(ix) • Is the 7.5 mmol/g/h the peak instantaneous rate or the slope of a linearized uptake curve?

We have now significantly revised the kinetics section (see reviewer 1, point 5). We show the kinetics data for PCS-OH alongside several literature materials. We no longer report the rate, and focus on a comparison of the raw data instead. From the revised SI:

Fig. S17. Direct air capture uptake kinetics measured by thermogravimetric analysis. In (a), data are shown for PCS-OH, 30 °C, 90 mL/min gas flow, 400 ppm CO₂ in air. In (b), data are shown for: PCS-OH (14 months after sample preparation, this work), 30 °C, 90 mL/min gas flow, 400 ppm CO₂ in air. Zn-(ZnOH)₄(bibta)₃ metal-organic framework, 27 °C, 50 mL/min gas flow, 395 ppm of CO₂, 21% O₂, N₂ balance.¹ PEI/MMO composite (PEI67/Mg0.55Al-O), 25 °C, 100 mL/min gas flow, 400 ppm CO₂ in N₂.² PEI/SBA-15 composite (PEI67/SBA-15), 25 °C, 100 mL/min gas flow, 400 ppm CO₂ in N₂.² These results highlight the excellent kinetic performance of PCS-OH.”

(x) • The discussion on page 8 does not discuss the role of RH, but it is mentioned in the figure caption. It is only on page 9 that the discussion of water begins. Perhaps these can be combined somehow?

Please see reviewer 1, point 10. We have now also made significant changes to the discussion of the impacts of relative humidity, and we have made the discussion more quantitative throughout. We believe this has improved this section, and hope this addresses the reviewers' suggestion.

(xi) • Does the water leave the sorbent during regeneration? Have moisture adsorption isotherms and subsequent desorption experiments been performed?

Fig. S20 shows the water adsorption and desorption isotherm at 25°C. The adsorption is reversible at room temperature, and we therefore anticipate complete removal of water at the elevated temperatures used for regeneration.

G. The references are appropriate.

H. The authors provide an adequate introduction. It is concise and on point.

Many thanks for these comments.

Reviewer Reports on the First Revision:

Referees' comments:

Referee #1 (Remarks to the Author):

Following this reviewer's comments the authors have made changes to the manuscript. However some points still need clarification/correction.

In Figure S17 graph (a) needs to be complemented or a new graph added showing the CO₂ uptake as a function of time for all the adsorbents selected in (b). This, at least in the range of 0-0.3 mmol/g CO₂ uptake of interest to compare with PCS-OH. This will allow for a more direct comparison of the kinetics, apples to apples. Otherwise, because all the adsorbents selected here have substantially different maximum CO₂ uptakes, figure (b) is not really adapted to compare kinetics in this case and is therefore not suitable to conclude that the kinetics of PCS-OH are better or "excellent".

In the conclusion and other places, the authors claim that the adsorbent presented here should enable an energy-efficient DAC process that requires renewable electricity as the only input. What are the facts to substantiate this claim? Have the authors estimated how much kWh/kgCO₂ captured will be needed? At the low CO₂ uptake measured under DAC conditions (0.08-0.2 mmol/g) it seems that the electrical energy needed per g or kg of CO₂ would be quite high and comparable if not higher than other processes. Electricity is needed to heat the entire charged sorbent to 130C for desorption. With a specific heat capacity of about 0.7 kJ/kg K for carbon, heating this assembly from rt to 130C requires about 70 kJ/kg. If only 0.08 mmol of CO₂ are captured by DAC under a realistic scenario with a RH of 38%, which as mentioned is pretty low, it means that only about 3.5 g of CO₂ would be captured by 1 kg of sorbent material in each cycle. This means that 70 kJ at a minimum would be required to desorb 3.5 g of CO₂ leading to $70 \times (1000/3.5) = 20000$ kJ/kg CO₂ captured, just for heating. This corresponds to about 5.5 kWh/kg of CO₂ captured just for resistive heating, not including any other energy need for the process. This should be compared with other estimates such as for example in "A Process for Capturing CO₂ from the Atmosphere", Keith, David W. et al. Joule, Volume 2, Issue 8, 1573 – 1594 as well as numerous other paper/reviews since then. In that case the process described here does not seem very efficient, at least in its present state.

Paragraph on line 242-249: How much CO₂ was actually captured in each consecutive cycles presented in Figure S18. Please add the data.

Referee #2 (Remarks to the Author):

The authors have made substantial changes to the manuscript, improving the data quality and also the

clarity with which the data are presented. In its revised form, the manuscript now more clearly highlights the novelty and importance of the work and it is recommended for publication. The reviewer congratulates the authors on a high quality project and great improvements to the manuscript from first submission.

Author Rebuttals to First Revision:

Referee #1 (Remarks to the Author):

Following this reviewer's comments the authors have made changes to the manuscript. However some points still need clarification/correction.

1. In Figure S17 graph (a) needs to be complemented or a new graph added showing the CO₂ uptake as a function of time for all the adsorbents selected in (b). This, at least in the range of 0-0.3 mmol/g CO₂ uptake of interest to compare with PCS-OH. This will allow for a more direct comparison of the kinetics, apples to apples. Otherwise, because all the adsorbents selected here have substantially different maximum CO₂ uptakes, figure (b) is not really adapted to compare kinetics in this case and is therefore not suitable to conclude that the kinetics of PCS-OH are better or “excellent”.

Thank you for this suggestion. The graph in Figure S17 has been updated as suggested, with a new part (b) inserted. As the reviewer suggests, this enables a better comparison of the kinetics of these materials, and shows that PCS-OH has comparable kinetics to several best-in-class sorbents. From the revised SI:

“

Fig. S17. Direct air capture uptake kinetics measured by thermogravimetric analysis. In (a), data are shown for PCS-OH, 30 °C, 90 mL/min gas flow, 400 ppm CO₂ in air. In (b), data are shown for: PCS-OH (14 months after sample preparation, this work), 30 °C, 90 mL/min gas flow, 400 ppm CO₂ in air. Zn-(ZnOH)₄(bibta)₃ metal-organic framework, 27 °C, 50 mL/min gas flow, 395 ppm of CO₂, 21% O₂, N₂ balance.¹ PEI/MMO composite (PEI67/Mg0.55Al-O), 25 °C, 100 mL/min gas flow, 400 ppm CO₂ in N₂.² PEI/SBA-15 composite (PEI67/SBA-15), 25 °C, 100 mL/min gas flow, 400 ppm CO₂ in N₂.² In (c), the data from (b) are plotted in terms of percentage uptake for each sorbent. The results in (b) show that PCS-OH can capture a comparable quantity of carbon dioxide per unit time to literature sorbents for times of approximately 10 to 15 minutes. Due to the rapid saturation of PCS-OH shown in (c), a relatively short adsorption cycle time should be used for this material.”

We have further revised our previous claim that the kinetic performance of PCS-OH is “excellent”. From the revised main text:

*“The DAC kinetics for **PCS-OH** are also promising, with a comparable CO₂ capture rate to several benchmark sorbents, albeit with a lower saturation capacity (Fig. S17).”*

2. In the conclusion and other places, the authors claim that the adsorbent presented here should enable an energy-efficient DAC process that requires renewable electricity as the only input. What are the facts to substantiate this claim? Have the authors estimated how much kWh/kgCO₂ captured will be needed? At the low CO₂ uptake measured under DAC conditions (0.08-0.2 mmol/g) it seems that the electrical energy needed per g or kg of CO₂ would be quite high and comparable if not higher than other processes. Electricity is needed to heat the entire charged sorbent to 130C for desorption. With a specific heat capacity of about 0.7 kJ/kg K for carbon, heating this assembly from rt to 130C requires about 70 kJ/kg. If only 0.08 mmol of CO₂ are captured by DAC under a realistic scenario with a RH of 38%, which as mentioned is pretty low, it means that only about 3.5 g of CO₂ would be captured by 1 kg of sorbent material in each cycle. This means that 70 kJ at a minimum would be required to desorb 3.5 g of CO₂ leading to $70 \times (1000/3.5) = 20000$ kJ/kg CO₂ captured, just for heating. This corresponds to about 5.5 kWh/kg of CO₂ captured just for resistive heating, not including any other energy need for the process. This should be compared with other estimates such as for example in “A Process for Capturing CO₂ from the Atmosphere”, Keith, David W. et al. Joule, Volume 2, Issue 8, 1573 – 1594 as well as numerous other paper/reviews since then. In that case the process described here does not seem very efficient, at least in its present state.

We thank the reviewer this detailed comment and suggestion. To address this we have first measured the experimental heat capacity for our material, PCS-OH, and we have added this to the supporting information:

“

Fig. S21. *Specific heat capacity measurements from differential scanning calorimetry for PCS-OH, using a DSC2500 from TA instruments. Data is shown for the sample heated from 25 °C to 90 °C after an initial heating and cooling cycle from 5 °C to 200 °C and back to 5°C. The mass of PCS-OH used was 2.7 mg. The average specific heat capacity between 25°C and 90°C is 0.62 J/(gK), and this value was used to estimate the minimum energy required for heating PCS-OH from 25°C to 90°C during regeneration, with a value of 40 J/g (see main text).*

Second, we have carried out the calculations described by the reviewer using the measured heat capacity and the experimental temperature swing of 25 to 90 °C, as used in the joule heating experiments. This gives energy consumption values of 6.5 and 11.4 GJ / ton CO₂ at relative humidities of 11 and 38%, respectively. We have included this analysis and some discussion in the revised manuscript.

“Using a desorption temperature of 90 °C and the low measured heat capacity of PCS-OH of 0.62 J/(gK) (Fig. S21), we estimate a minimum electrical energy consumption for sorbent heating of 6.5 and 11.4 GJ/ton CO₂ captured for RH values of 11 and 38%, respectively (equivalent to 1800 and 3200 kWh/ton CO₂). These values are comparable to those reported for a range of DAC processes.³² For example, when using aqueous KOH solutions the capture of 1 ton of CO₂ required either 8.8 GJ natural gas or 5.3 GJ natural gas and 77 kWh electricity in limiting cases.⁶”

While the energy consumption value is currently higher for charged-sorbents compared to state-of-the-art processes, the possibility to solely use electricity to run the process is an advantage. In addition, there is a straightforward method to improve the energy-efficiency of our process: increase the capacity of the charged sorbents. We thus propose the development of charged-sorbents with increased capacities as a promising future direction:

“A key advantage of charged-sorbents is that full electrification of the DAC process is possible, thereby avoiding issues with natural gas use and leakage, which can offset a significant fraction of the captured CO₂ in traditional DAC processes due to the high global warming potential of methane.³³ While this may justify the higher operating energy costs for charged-sorbents, future work should be carried out to improve their energy efficiencies. The most straightforward way to achieve this is to increase the sorbent CO₂ capacities, something that we are working towards in our laboratories.”

Finally, we have revised the sentence in the conclusion mentioned by the reviewer, which now reads:

“The promising oxidative stabilities combined with rapid Joule heating regeneration should enable a DAC process that requires renewable electricity as the only input.”

3. Paragraph on line 242-249: How much CO₂ was actually captured in each consecutive cycles presented in Figure S18. Please add the data.

Figure S18 and its caption have been revised to add this information. From the revised SI:

“

Fig. S18. DAC cycling experiments for PCS-OH in the sealed chamber, with 20 min Joule heating regeneration under nitrogen between cycles. The RH during DAC tests was controlled by a desiccant (silica gel) at 11% at 25 °C. The mass of PCS-OH used was 30 mg. The CO₂ capacities in these tests are listed in the figure, and correspond to gravimetric capacities of 0.11, 0.10, 0.10, 0.11, and 0.11 mmol g⁻¹, respectively.”

Referee #2 (Remarks to the Author):

The authors have made substantial changes to the manuscript, improving the data quality and also the clarity with which the data are presented. In its revised form, the manuscript now more clearly highlights the novelty and importance of the work and it is recommended for publication. The reviewer congratulates the authors on a high quality project and great improvements to the manuscript from first submission.

Reviewer Reports on the Second Revision:

Referees' comments:

Referee #1 (Remarks to the Author):

The authors have satisfactorily answered majority of this referee's queries. The manuscript is now publishable.